**DOI: 10.1038/ncomms14158**　　**OPEN**

# Potent single-domain antibodies that arrest respiratory syncytial virus fusion protein in its prefusion state

Iebe Rossey[1,2,*], Morgan S.A. Gilman[3,*], Stephanie C. Kabeche[3], Koen Sedeyn[1,2], Daniel Wrapp[3], Masaru Kanekiyo[4], Man Chen[4], Vicente Mas[5], Jan Spitaels[1,2], José A. Melero[5], Barney S. Graham[4], Bert Schepens[1,2], Jason S. McLellan[3] & Xavier Saelens[1,2]

Human respiratory syncytial virus (RSV) is the main cause of lower respiratory tract infections in young children. The RSV fusion protein (F) is highly conserved and is the only viral membrane protein that is essential for infection. The prefusion conformation of RSV F is considered the most relevant target for antiviral strategies because it is the fusion-competent form of the protein and the primary target of neutralizing activity present in human serum. Here, we describe two llama-derived single-domain antibodies (VHHs) that have potent RSV-neutralizing activity and bind selectively to prefusion RSV F with picomolar affinity. Crystal structures of these VHHs in complex with prefusion F show that they recognize a conserved cavity formed by two F protomers. In addition, the VHHs prevent RSV replication and lung infiltration of inflammatory monocytes and T cells in RSV-challenged mice. These prefusion F-specific VHHs represent promising antiviral agents against RSV.

[1] Medical Biotechnology Center, VIB, Technologiepark 927, Ghent B-9052, Belgium. [2] Department of Biomedical Molecular Biology, Ghent University, Ghent B-9052, Belgium. [3] Department of Biochemistry and Cell Biology, Geisel School of Medicine at Dartmouth, Hanover, New Hampshire 03755-3844, USA. [4] Vaccine Research Center, National Institute of Allergy and Infectious Diseases, National Institutes of Health, Bethesda, Maryland 20892, USA. [5] Centro Nacional de Microbiología and CIBER de Enfermedades Respiratorias, Instituto de Salud Carlos III, Majadahonda, 28220 Madrid, Spain. * These authors contributed equally to this work. Correspondence and requests for materials should be addressed to B.S. (email: Bert.Schepens@vib-ugent.be) or to J.S.M. (email: Jason.S.McLellan@dartmouth.edu) or to X.S. (email: xavier.saelens@vib-ugent.be).

Human respiratory syncytial virus (RSV) is the leading cause of lower respiratory tract infections in children under the age of five throughout the world. It is estimated that RSV infects about 33.8 million children in this age group annually, of which more than 3 million require hospitalization due to severe bronchiolitis or pneumonia[1]. Reinfections occur regularly throughout life because natural infection offers only limited immunity[2]. RSV is also recognized as a major pathogen for the elderly, with a disease burden similar to that of seasonal influenza[3]. Thus, there is an urgent need for therapeutics that can reduce disease caused by RSV.

Despite its medical importance and decades of intense research, there is still no licensed RSV vaccine nor an effective antiviral. The humanized monoclonal antibody (mAb) palivizumab (Synagis) reduces hospitalizations when administered prophylactically, but its high cost and limited efficacy restrict its use to high-risk infants[4]. Palivizumab neutralizes RSV by binding to the fusion (F) protein and preventing fusion of the viral membrane with the host-cell membrane[5]. RSV F is a class I fusion protein that is expressed as an inactive precursor, $F_0$, which is cleaved at two sites by a furin-like protease, leading to the formation of the disulfide-linked $F_2$ (N-terminal) and $F_1$ (C-terminal) subunits, which associate and trimerize to form the mature prefusion F protein[6]. Upon triggering, prefusion F partially refolds and inserts its hydrophobic fusion peptide into the membrane of the target cell. Fusion of the viral and host-cell membranes is facilitated by further refolding of the F protein into the stable postfusion conformation. Small molecules that bind to RSV F and prevent its structural remodelling, or F-specific antibodies that interfere with membrane fusion, can block RSV infection[7–10]. Such compounds are being clinically developed.

Palivizumab binds to antigenic site II on RSV F, which is one of two well-characterized antigenic sites that are present on both the pre- and postfusion conformations. However, intensive screening for human mAbs that potently neutralize RSV has resulted in the isolation of prefusion F-specific antibodies with more robust neutralizing activity than palivizumab[9,10]. Recently, RSV F was successfully stabilized in its prefusion conformation through the introduction of an intraprotomeric disulfide bond, cavity-filling mutations and a trimerization motif. This reagent, called DS-Cav1, has been instrumental in revealing that the vast majority of RSV-neutralizing immunoglobulins in human sera selectively bind to F in its prefusion conformation[11–13].

In addition to conventional antibodies, heavy-chain-only antibodies also exist in nature, for example, in both camelids and sharks[14,15]. The isolated antigen-recognition domains of these unusual antibodies are known as single-domain antibodies (VHHs). VHHs are very well suited for the development of therapeutics because of their small size, ease of production and physical stability that allows alternative routes of administration such as pulmonary delivery by nebulization[16]. A number of clinical trials are already ongoing with recombinant VHHs for the treatment of rheumatoid arthritis, cancer and infectious diseases[17–19]. ALX-0171 is an RSV-neutralizing VHH that binds to an epitope on RSV F that is similar to that of palivizumab[19]. In a phase I/IIa trial, hospitalized RSV-infected children were treated daily for three consecutive days with ALX-0171 delivered by an inhalation device[16]. The treatment was safe and did not lead to any treatment-related serious adverse events. Interestingly, the study also revealed a trend towards a therapeutic effect, based on reduced viral loads in nasal swabs and clinical symptoms. In contrast, a similar trial with motavizumab—an affinity matured version of palivizumab—did not alter viral replication or improve clinical symptoms when administered after infection[20]. This different outcome might be explained by the direct delivery of ALX-0171 to the lungs whereas only about 0.2% of systemically administered antibody ends up in the lung lumen[21].

We hypothesized that a prefusion-specific VHH would have a much stronger antiviral effect than a conformation-independent VHH like ALX-0171. Here, we present the isolation and characterization of two llama-derived VHHs that potently neutralize RSV A and B subtypes and selectively bind prefusion F with picomolar affinity. Structural studies reveal that these two VHHs bind to a conserved quaternary epitope composed of two F protomers. Furthermore, prophylactic treatment of mice with these VHHs potently controlled RSV infection and associated inflammation.

## Results

**Isolation of VHHs with exceptional RSV-neutralizing activity.** To obtain VHHs that selectively bind to the prefusion conformation of RSV F, a llama was immunized with recombinant F protein that was stabilized in the prefusion conformation, that is, DS-Cav1 (ref. 13). Plasma prepared after six weekly immunizations with prefusion F had potent RSV subtype A and B neutralizing activity (Supplementary Fig. 1). Peripheral blood mononuclear cells were isolated from the prefusion F-immunized llama and used to generate a VHH phage-display library. One round of panning on immobilized prefusion F protein resulted in a 140-fold enrichment in candidate prefusion F-binding phages. In a first approach to select RSV-neutralizing VHHs, we randomly selected 90 clones from this enriched population and tested binding to RSV F protein by ELISA. Thirty-seven clones expressed VHH that bound to either pre- or postfusion F protein (FΔFP)[22]; twenty-eight of these had a unique sequence and were subsequently expressed in *Pichia pastoris*. RSV A2-neutralizing activity was present in the crude growth medium of *P. pastoris* clones 1, 4, 8, 13 and 44, which were retained for further analysis (Supplementary Fig. 2a).

In an alternative approach, the VHH cDNA inserts obtained after one round of panning on prefusion F were cloned as a pool into the *P. pastoris* expression vector and the resulting library was used to transform *P. pastoris*. The crude culture growth medium of individual *P. pastoris* transformants was then tested for RSV-neutralizing activity. One candidate, named F-VHH-L66, displayed clear RSV A2-neutralizing activity (Supplementary Fig. 2b). We purified the VHHs from the culture supernatant of the selected *P. pastoris* transformants for further analysis. All six purified VHHs (F-VHH-1, -4, -8, -13, -44 and -L66) neutralized RSV A and RSV B subtypes *in vitro*, with F-VHH-4 and F-VHH-L66 exhibiting the most potent neutralization, with an $IC_{50}$ value below 0.1 nM for both RSV strains (Fig. 1a,b). These two VHHs were therefore selected for further characterization. Sequence analysis of F-VHH-4 and -L66 revealed an identical complementarity-determining region (CDR) 1, a similar CDR2, but a very different CDR3 (Fig. 1c and Supplementary Fig. 3). F-VHH-4 and -L66 both have a cysteine residue at position 100c in the CDR3 and at position 50 in the CDR2. A disulfide bond between the CDR3 and a residue at position 33 or 50 is a hallmark of VHHs and helps to stabilize the extended CDR3 loop conformation[23].

To assess the antiviral potency of the VHHs in more detail, RSV-neutralization assays were performed in which F-VHH-4 and -L66 were compared with several F-specific mAbs (palivizumab, motavizumab, D25 and AM22)[9,24,25]. Remarkably, despite their monovalent character, F-VHH-4 and -L66 neutralized RSV A and B with picomolar $IC_{50}$ values, which are similar to or lower than those for the bivalent prefusion

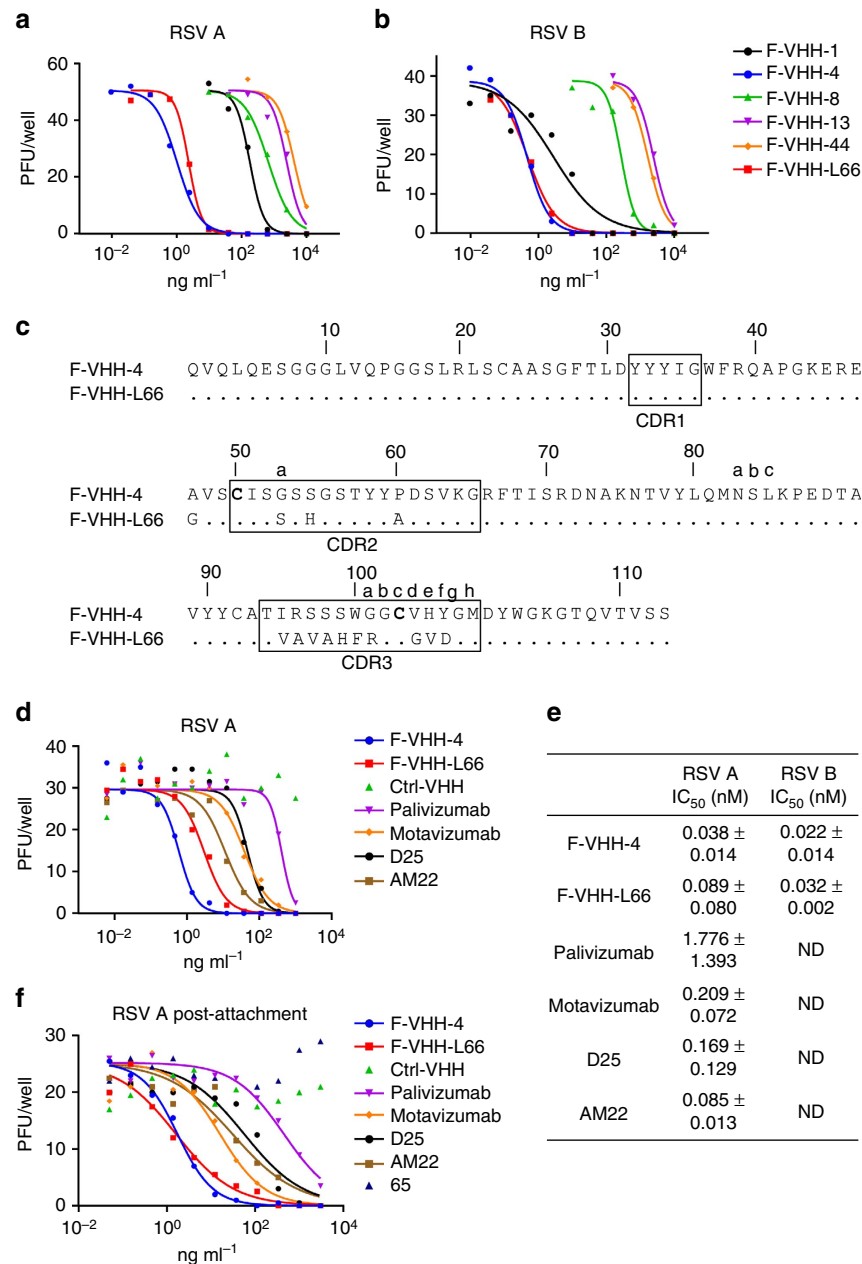

**Figure 1 | F-VHHs with potent RSV-neutralizing activity.** RSV-neutralizing activity by F-VHHs. (**a**) RSV A2 (50 pfu per well) or (**b**) RSV B49 (40 pfu per well) was preincubated with different concentrations of VHH before infection of Vero cells. Three days later, the viral plaques were stained with polyclonal anti-RSV serum. (**c**) Amino acid sequences of F-VHH-4 and F-VHH-L66 with Kabat numbering. The complementarity-determining regions (CDRs) are boxed. Cysteines in CDR2 and CDR3 are in bold. (**d**) RSV A2 (30 pfu per well) plaque-reduction activity of F-VHH-4 and F-VHH-L66 compared with mAbs. The Ctrl-VHH is specific for an irrelevant target. (**e**) IC50 values of F-VHHs and mAbs against RSV A2 and RSV B49 as determined by plaque-reduction assay. Mean values ± s.d. from four (VHHs), three (D25) or two (palivizumab, motavizumab and AM22) repeat experiments are depicted for RSV A. Mean values ± s.d. from two experiments are depicted for RSV B. ND: Not determined. (**f**) Dilution series of VHHs and mAbs were added to HEp-2 cells that had been preincubated with RSV A2 (25 pfu per well) at 4 °C for 2 h, allowing viral attachment. After 2 h at 37 °C, the VHHs or antibodies were washed away and infection was allowed during two days at 37 °C after which the plaques were stained. Antibody, 65 is a mouse IgG2a monoclonal antibody that is specific for an irrelevant antigen.

F-specific mAbs D25 and AM22 (Fig. 1d,e). The VHHs were also tested in a neutralization assay against a set of clinical RSV isolates and a set of molecular clones built on the RSV Line 19 backbone, but with F and G derived from primary clinical isolates. These strains were as susceptible as the lab strains to neutralization by the VHHs (Supplementary Table 1 and Supplementary Fig. 4).

In HEp-2 cells the F protein is involved in both viral attachment and membrane fusion[26–30]. We therefore investigated if the VHHs could prevent infection after attachment of viral particles to the cells. Similar to palivizumab and D25, F-VHH-4 and F-VHH-L66 prevented infection when added after virion attachment (Fig. 1f). However, mixing of RSV with the neutralizing VHHs before infection did not hinder viral attachment to HEp-2 target cells, indicating that F-VHH-4 and -L66 inhibit RSV *in vitro* by blocking membrane fusion post-attachment (Supplementary Fig. 5).

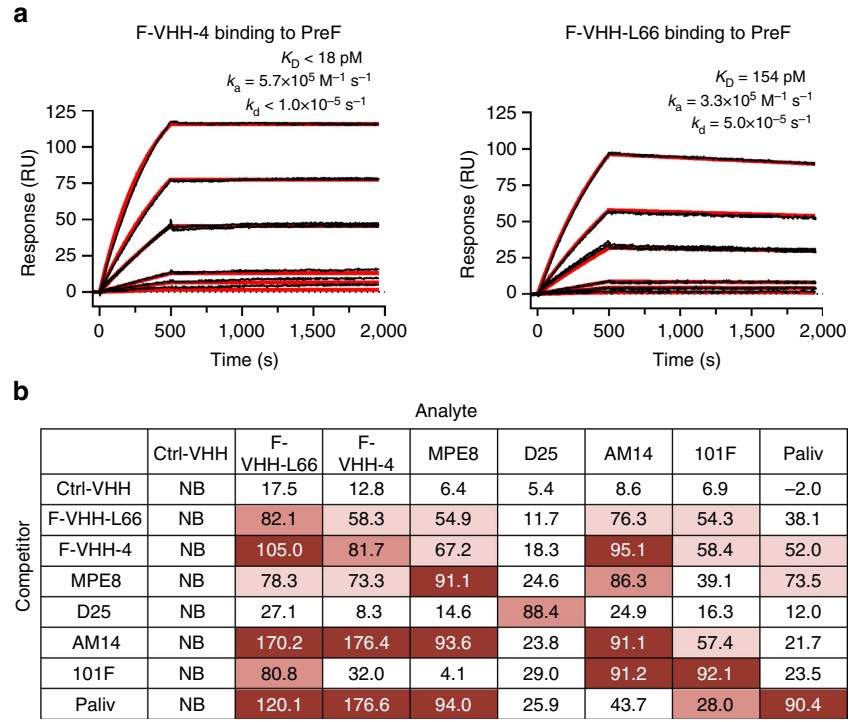

**a**

F-VHH-4 binding to PreF

$K_D < 18$ pM
$k_a = 5.7 \times 10^5$ M$^{-1}$ s$^{-1}$
$k_d < 1.0 \times 10^{-5}$ s$^{-1}$

F-VHH-L66 binding to PreF

$K_D = 154$ pM
$k_a = 3.3 \times 10^5$ M$^{-1}$ s$^{-1}$
$k_d = 5.0 \times 10^{-5}$ s$^{-1}$

**b**

| | | | | Analyte | | | | |
|---|---|---|---|---|---|---|---|---|
| | | Ctrl-VHH | F-VHH-L66 | F-VHH-4 | MPE8 | D25 | AM14 | 101F | Paliv |
| Competitor | Ctrl-VHH | NB | 17.5 | 12.8 | 6.4 | 5.4 | 8.6 | 6.9 | −2.0 |
| | F-VHH-L66 | NB | 82.1 | 58.3 | 54.9 | 11.7 | 76.3 | 54.3 | 38.1 |
| | F-VHH-4 | NB | 105.0 | 81.7 | 67.2 | 18.3 | 95.1 | 58.4 | 52.0 |
| | MPE8 | NB | 78.3 | 73.3 | 91.1 | 24.6 | 86.3 | 39.1 | 73.5 |
| | D25 | NB | 27.1 | 8.3 | 14.6 | 88.4 | 24.9 | 16.3 | 12.0 |
| | AM14 | NB | 170.2 | 176.4 | 93.6 | 23.8 | 91.1 | 57.4 | 21.7 |
| | 101F | NB | 80.8 | 32.0 | 4.1 | 29.0 | 91.2 | 92.1 | 23.5 |
| | Paliv | NB | 120.1 | 176.6 | 94.0 | 25.9 | 43.7 | 28.0 | 90.4 |

**Figure 2 | F-VHH-4 and -L66 bind a similar epitope on prefusion F that partially overlaps with four previously characterized epitopes.** (**a**) Surface plasmon resonance (SPR) sensorgrams for the binding of F-VHH-4 (left) and F-VHH-L66 (right) to immobilized prefusion F. A buffer-only sample was injected over the prefusion F and reference flow cells, followed by 2-fold serial dilutions of F-VHH-4 or F-VHH-L66 ranging from 5 nM to 39.1 pM, with a duplication of the 1.25 nM concentration. The data were double-reference subtracted and fit to a 1:1 binding model (red lines). (**b**) Prefusion F protein was immobilized on AR2G biosensors. The biosensors were dipped in competitor antibodies followed by analyte antibodies. Per cent inhibitions were defined by comparing binding maxima of the analyte antibody in the absence and presence of each competitor. NB, no binding.

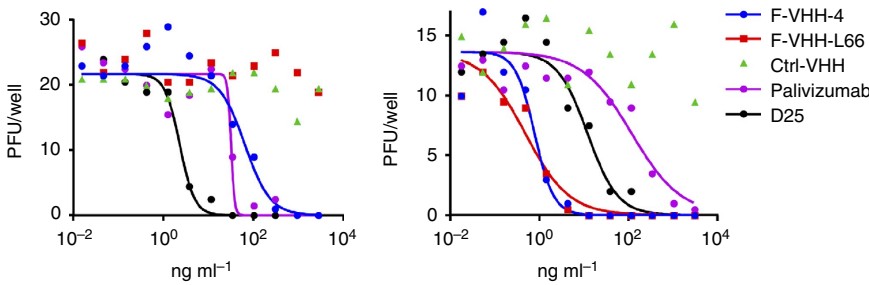

**Figure 3 | Thr50Asn substitution in F$_2$ enables viral escape from F-VHH-4 and F-VHH-L66 neutralization.** F-VHH-4 escape virus (left panel) or RSV A2 (right panel) (25 PFU) was incubated for 30 min at 37 °C with different concentrations of the indicated VHH or mAb before addition to a monolayer of Vero cells. Three days later, cells were fixed and stained with polyclonal anti-RSV serum to visualize the plaques (enumerated in the Y axis).

**VHHs bind a unique prefusion F-specific epitope**. To investigate whether the neutralizing VHHs specifically recognize the prefusion conformation of RSV F, the binding kinetics to recombinant pre- and postfusion F proteins were measured by surface plasmon resonance (SPR). The VHHs bound tightly to prefusion-stabilized F, whereas no binding to postfusion F could be detected (Fig. 2a and Supplementary Fig. 6). F-VHH-4 bound to immobilized prefusion RSV F with an equilibrium dissociation constant ($K_D$) lower than 18 pM. The off-rate was extremely slow and outside the range measurable by SPR, therefore, the upper limit of the $K_D$ was estimated using the measured association rate constant ($5.7 \times 10^5$ M$^{-1}$ s$^{-1}$) and the lower limit of detection for the dissociation rate constant ($1 \times 10^{-5}$ s$^{-1}$). F-VHH-L66 also bound to prefusion F with sub-nanomolar affinity. However, compared with F-VHH-4, the F-VHH-L66 association rate constant was lower

($3.3 \times 10^5$ M$^{-1}$ s$^{-1}$) and the dissociation rate constant was higher ($5 \times 10^{-5}$ s$^{-1}$), resulting in a $K_D$ of 154 pM. Binding of the VHHs to cells transfected with a codon-optimized F-expression plasmid and to RSV-infected cells was also observed (Supplementary Figs 7 and 8), demonstrating that the VHHs recognize full-length F in the context of biological membranes.

To gain more insight into the epitopes that are targeted by F-VHH-4 and -L66, we performed cross-competition binding experiments using biolayer interferometry and a set of mAbs with known epitopes. This analysis revealed that the two VHHs competed with each other for binding to immobilized prefusion F and thus likely bound to overlapping epitopes (Fig. 2b). Binding by D25, which recognizes site Ø on prefusion F, was not hindered by F-VHH-4 or -L66, which suggests that neither VHH bound to this site. In contrast, palivizumab,

**Table 1 | Crystallographic data collection and refinement statistics.**

|  | F-VHH-4 | F-VHH-4-prefusion F | F-VHH-L66-prefusion F |
|---|---|---|---|
| PDB ID | 5TP3 | 5TOJ | 5TOK |
| *Data collection* |  |  |  |
| Space group | $P2_12_12_1$ | $P3_221$ | $P3_121$ |
| Cell dimensions |  |  |  |
| $a, b, c$ (Å) | 47.7, 47.8, 149.8 | 173.2, 173.2, 153.3 | 138.9, 139.9, 221.9 |
| $\alpha, \beta, \gamma$ (°) | 90, 90, 90 | 90, 90, 120 | 90, 90, 120 |
| Resolution (Å) | 34.5–1.9 | 38.0–3.3 | 50.4–3.8 |
|  | (1.92–1.87) | (3.43–3.30) | (4.06–3.80) |
| $R_{merge}$ | 0.118 (1.125) | 0.188 (1.081) | 0.332 (1.246) |
| $I/\sigma I$ | 11.1 (2.1) | 9.2 (2.1) | 5.2 (1.5) |
| $CC_{1/2}$ | 0.995 (0.465) | 0.995 (0.545) | 0.975 (0.549) |
| Completeness (%) | 100 (100) | 99.9 (100) | 99.9 (100) |
| Redundancy | 6.6 (5.6) | 7.4 (7.5) | 5.3 (5.5) |
|  |  |  |  |
| *Refinement* |  |  |  |
| Resolution (Å) | 34.5–1.9 | 38.0–3.3 | 50.4–3.8 |
|  | (1.94–1.87) | (3.38–3.30) | (3.95–3.80) |
| Unique reflections | 28,958 (2,824) | 40,277 (2,821) | 24,981 (2,726) |
| $R_{work}/R_{free}$ (%) | 17.0/20.8 | 18.4/23.4 | 24.8/28.6 |
| No. atoms |  |  |  |
| Protein | 1,910 | 14,082 | 13,952 |
| Glycan (NAG) | 0 | 42 | 42 |
| Ion ($PO_4$) | 0 | 0 | 5 |
| Water | 143 | 0 | 0 |
| B-factors |  |  |  |
| Protein | 25.7 | 94.5 | 124.0 |
| Glycan (NAG) | — | 97.3 | 117.9 |
| Ion ($PO_4$) | — | — | 149.0 |
| Water | 29.4 | — | — |
| R.m.s. deviations |  |  |  |
| Bond lengths (Å) | 0.007 | 0.005 | 0.007 |
| Bond angles (°) | 0.81 | 1.20 | 1.13 |
| Ramachandran |  |  |  |
| Favoured (%) | 97.2 | 96.5 | 96.6 |
| Allowed (%) | 2.9 | 3.3 | 3.0 |
| Outliers (%) | 0 | 0.2 | 0.4 |

Values in parentheses are for the highest-resolution shell.

AM14, MPE8 and 101F interfered with binding of the two VHHs to prefusion F and vice versa, suggesting that the VHH epitope is located in the vicinity of the binding sites of these four antibodies, approximately midway between the apex of the trimer and the viral membrane.

To further narrow down the binding site of the VHHs, we selected for viruses resistant to F-VHH-4 and -L66. Two rounds of selection in the presence of F-VHH-L66 and four rounds of selection in the presence of F-VHH-4 led to the isolation of a single type of escape variant that had a threonine to asparagine substitution at position 50 in the $F_2$ subunit. F-VHH-L66 failed to neutralize this mutant virus, whereas this virus was still susceptible to inhibition by F-VHH-4, although this required 300-fold more F-VHH-4 compared with the parental virus (Fig. 3). Thr50 resides at the floor of a cavity in prefusion F that is surrounded by the epitopes of palivizumab, AM14 and 101F[5,31,32], suggesting that both VHHs bind to this cavity.

**F-VHHs bind a cavity formed by two F protomers.** To precisely define the epitopes recognized by F-VHH-4 and -L66, their crystal structures in complex with prefusion F were determined to 3.3 and 3.8 Å resolution, respectively (Table 1). In addition, the crystal structure of unbound F-VHH-4 was determined to

1.9 Å resolution (Table 1 and Supplementary Fig. 9). The complex structures revealed that despite substantial differences in CDR3 sequence, F-VHH-4 and -L66 bind in a nearly identical manner to a cavity formed by the boundary of two F protomers (Fig. 4a). This cavity is bordered by antigenic site II of one protomer and antigenic site IV of the neighbouring protomer, and the location of this epitope between the binding sites of palivizumab, 101F and AM14 explains the ability of the VHHs to compete with each of these antibodies (Fig. 2b and Supplementary Fig. 10). This epitope has also been mapped by mutagenesis as the likely epitope for MPE8, which cross-neutralizes RSV, human metapneumovirus (hMPV) and pneumonia virus of mice (PVM)[10]. The VHHs, however, while being broadly neutralizing for RSV, did not neutralize hMPV (Supplementary Table 2).

F-VHH-4 and -L66 each bury nearly 1,200 Å$^2$ on prefusion F, with approximately 60% of the total buried surface area on one protomer and 40% on the adjacent protomer (Fig. 4b). The distribution of the VHH epitopes among the two protomers suggested that F-VHH-4 and -L66 may be trimer-specific, which is a property that thus far has only been described for antibody AM14 (ref. 31). To confirm the trimer-specificity of the VHHs, their binding kinetics for monomeric prefusion F were measured by SPR (Supplementary Fig. 11). The $K_D$ of F-VHH-4 binding to monomeric prefusion F was 23 nM, which is over

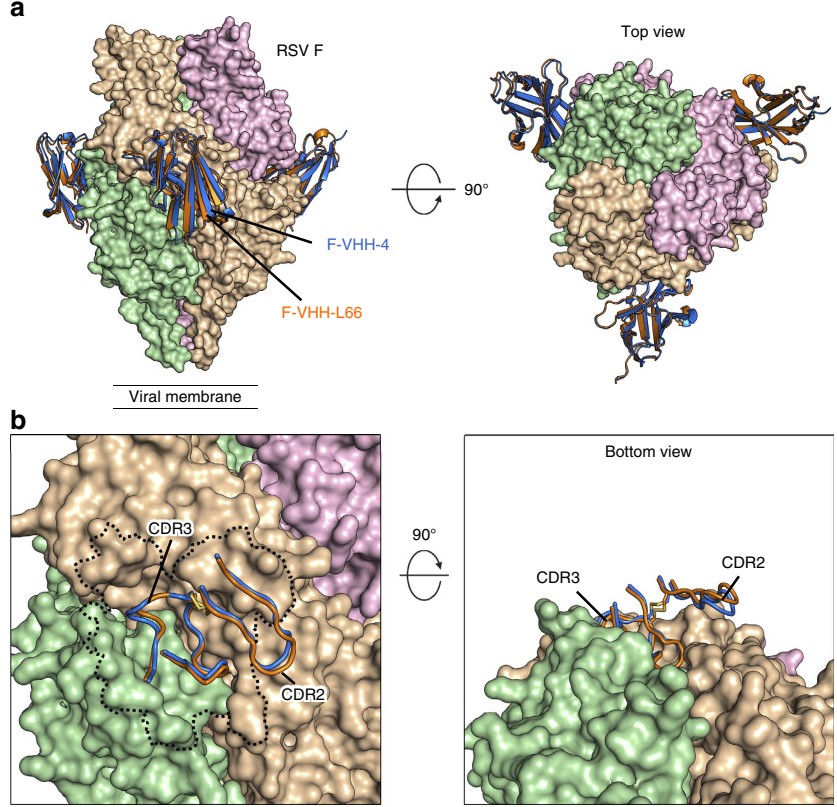

**Figure 4 | F-VHHs bind an epitope formed by two protomers of prefusion F.** (**a**) Superposition of the crystal structures of three F-VHH-4 s (blue) and three F-VHH-L66s (orange) bound to prefusion RSV F, viewed from the side (left) and the top (right). (**b**) Close-up of the VHH-binding cavity, viewed from the side (left) and from the bottom (right). The VHH epitope is outlined with a dotted black line in the side view. The CDR2s and CDR3s of both VHHs are shown as tubes.

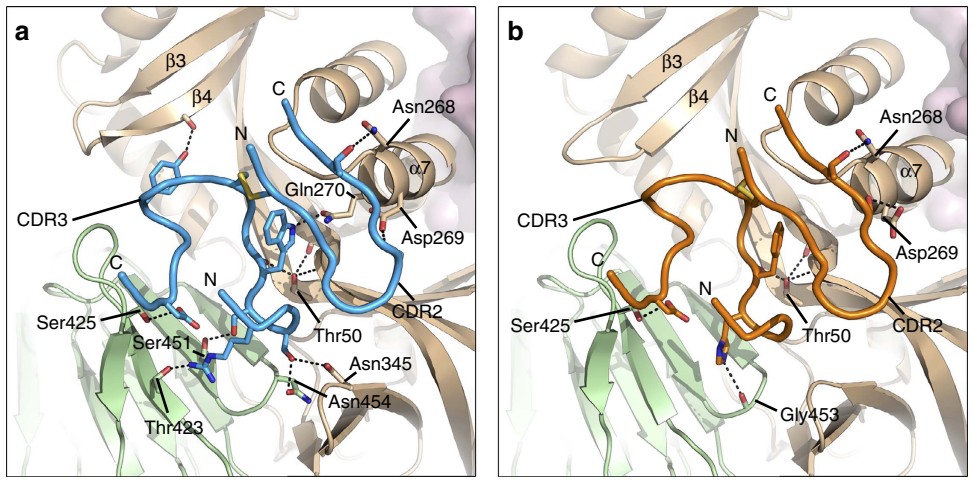

**Figure 5 | F-VHHs make direct contacts with residues on two neighbouring protomers of prefusion F.** Close-up of the CDR3 and CDR2 from (**a**) F-VHH-4 and (**b**) F-VHH-L66 bound to prefusion F. CDR2 and CDR3 are shown as tubes with the N- and C-termini labelled. Side-chains and main-chain involved in hydrogen bonding are shown as sticks, with hydrogen bonds depicted as black dotted lines. Residues of the F protein are labelled.

1,000-fold weaker than the affinity measured for trimeric prefusion F. No binding of F-VHH-L66 to monomeric prefusion F was detected, even at the highest concentration tested (500 nM) (Supplementary Fig. 11). Both VHHs were also able to bind and trap wild-type F protein ectodomain in the trimeric prefusion conformation (Supplementary Fig. 12). Therefore, both VHHs have a high selectivity for the trimeric state of prefusion F.

The CDR3s of F-VHH-4 and -L66 are submerged in the interprotomeric cavity and therefore mediate the majority

of the interactions with RSV F (Fig. 5 and Supplementary Table 3). The main chain of both F-VHH-4 and -L66 forms a hydrogen bond with Thr50 of $F_2$, the residue mutated in the escape viruses isolated for both VHHs. A number of hydrogen bonds are also formed between the CDR3 and the antiparallel β-strands that neighbour antigenic site IV. The CDR3 of each VHH is disulfide-linked to CDR2, which hydrogen bonds with the α7 helix of antigenic site II (Fig. 5). Upon membrane fusion, the domains of each RSV F protomer contacted by the

VHHs undergo a rigid-body movement that causes their relative positions to change substantially between the prefusion and postfusion states. Superposition of the postfusion F structure onto the VHH-bound prefusion structures reveals a significant clash between the loop connecting α6 and α7 of postfusion F and the CDR3 of either VHH (Supplementary Fig. 13). Therefore, our structural data suggest that F-VHH-4 and -L66 neutralize RSV and block fusion by preventing the movement of domains required for conversion of prefusion RSV F to the postfusion state. In addition, both VHHs bury approximately $100\,\text{Å}^2$ on the β4 strand, which dramatically rearranges to become a portion of the α5 helix of postfusion F. Although the buried surface area in this region is relatively small, these contacts may also contribute to the prefusion specificity of the VHHs by preventing the refolding of β4 during the transition from prefusion to postfusion F.

**F-VHHs reduce replication of RSV *in vivo*.** In a final set of experiments, we evaluated the protective potential of F-VHH-4 and -L66 *in vivo*. Both F-VHHs strongly reduced virus replication when administered intranasally to BALB/c mice at a dose of $30\,\mu\text{g}$ ($1.5\,\text{mg}\,\text{kg}^{-1}$) before challenge with RSV (Supplementary Fig. 14). We selected F-VHH-4 for subsequent animal experiments because it has a slightly higher neutralizing activity than F-VHH-L66. Intranasal administration of $10\,\mu\text{g}$ ($0.5\,\text{mg}\,\text{kg}^{-1}$) F-VHH-4 or palivizumab to BALB/c mice 4 h before challenge with $1\times10^6$ PFU of RSV A2 resulted in undetectable levels of virus in the lungs and the bronchoalveolar lavage on day five after challenge (Fig. 6a and Supplementary Fig. 15). This apparent lack of virus in the lungs of F-VHH-4- or palivizumab-treated mice could be due to an *in vitro* effect of the remaining VHH or antibody in the lung homogenates. Therefore, viral RNA was quantified by an RSV-specific RT-qPCR. The lungs of mice treated with F-VHH-4 were negative or borderline positive for RSV RNA, whereas the relative amount of RSV RNA indicated at least a 170-fold higher viral load in samples from palivizumab-treated mice compared with F-VHH-4-treated mice (Fig. 6b). Finally, we also assessed the effect of F-VHH-4 treatment on the influx of immune cells in the lungs of RSV A2-challenged mice five days after infection. In contrast to mice treated with Ctrl-VHH before challenge, mice treated with F-VHH-4 or palivizumab had significantly less alveolar influx of dendritic cells, monocytes, and CD8$^+$ and CD4$^+$ T lymphocytes (Fig. 6c). Lowering the amount of F-VHH-4 to $1\,\mu\text{g}$ per mouse resulted in the same protective effect (Supplementary Fig. 16).

**Discussion**
Our aim was to obtain molecules with very high RSV-neutralizing activity that could be developed as a therapeutic for the treatment of RSV infections. We describe a new class of VHHs that bind specifically to the prefusion conformation of the RSV F protein. Such VHHs were obtained by the immunization of a llama with a soluble prefusion-stabilized F protein combined with an RSV-neutralization-based screening strategy. The resulting VHHs display broad RSV-neutralizing activity, bind specifically and with high affinity to the prefusion conformation of F and inhibit RSV replication *in vivo*. The potent RSV-neutralizing activity of the F-VHHs correlates with their ultra-high affinity for the prefusion F protein. By tightly binding to an epitope that is only present on the trimeric prefusion conformation, the VHHs presumably interfere with the transition of the F protein from the prefusion to the postfusion state, thereby preventing fusion with the host cell.

F-VHH-4 and F-VHH-L66 have only 5 out of 15 amino acids in common in their CDR3. Yet, F-VHH-4 and -L66 bind

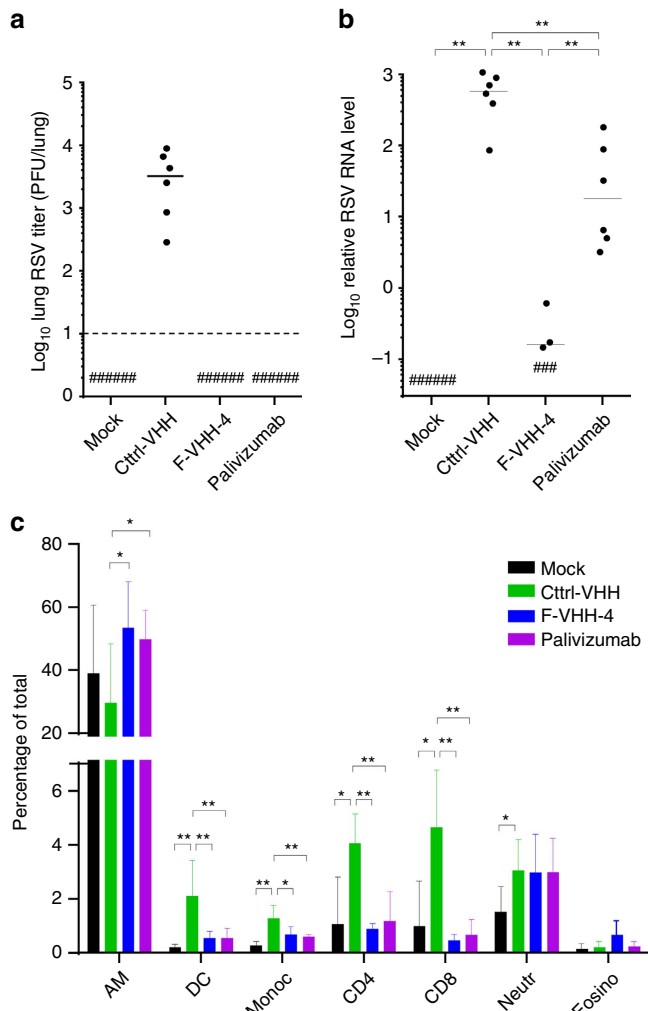

**Figure 6 | F-VHH-4 inhibits RSV A2 replication in mice.** Groups of six BALB/c mice were treated intranasally with $0.5\,\text{mg}\,\text{kg}^{-1}$ of F-VHH-4, palivizumab or Ctrl-VHH 4 h before infection with $1\times10^6$ RSV A2 or PBS (mock). Five days after infection, the pulmonary RSV load was determined by (**a**) plaque assay (dashed line represents the detection limit) or (**b**) by quantification of the amount of lung viral RNA using RT-qPCR. Horizontal lines indicate medians. $^*P<0.05$, $^{**}P<0.01$ (Mann–Whitney $U$-test). (**c**) Five days after infection, BAL fluid was prepared and the percentage of immune cells was determined by flow cytometry. Bars represent the average percentage of the indicated cell type ± s.d. ($n=6$) (AM, alveolar macrophages; DC, dendritic cells; Monoc, monocytes; CD4, CD4$^+$ T cells; CD8, CD8$^+$ T cells; Neutr, neutrophils; Eosino, eosinophils). $^*P<0.05$, $^{**}P<0.01$ (Mann–Whitney $U$-test).

closely overlapping epitopes in a nearly identical way. On the nucleotide level, the VHHs are 94% identical, indicating that they probably emerged from the same recombined germline sequences. If so, it is remarkable that two very different paths of affinity maturation converge to highly similar modes of binding. Crystallization studies showed that the CDR3 of both VHHs bind into a cavity formed by two protomers. This illustrates the tendency of VHHs to preferably bind clefts or cavities, whereas conventional antibodies in general bind either flat surfaces or extruding epitopes[33]. The epitope of F-VHH-4 and F-VHH-L66 overlaps with the epitope mapped by mutagenesis for the broadly neutralizing antibody MPE8, which has neutralizing activity against human and bovine

RSV, hMPV and PVM[10]. Nevertheless, the VHHs do not neutralize hMPV. RSV and hMPV F proteins have only 33% overall amino acid sequence identity[34], yet a substantial area of the surmised MPE8 epitope is conserved between the two viruses. Although the VHHs bind to a similar region, they make crucial contacts with a number of residues that are not conserved between RSV F and hMPV F, explaining the lack of hMPV neutralization by the F-VHHs. Thus, in this case there appears to be a trade-off between the potency of the VHHs and the breadth of MPE8.

We addressed the *in vitro* escape potential of RSV in the presence of the VHHs. F-VHH-4 and F-VHH-L66 escape viruses had a single mutation resulting in a substitution of Thr50, which is located at the bottom of the cavity bound by the VHHs. Although resistant to F-VHH-L66, this virus could still be neutralized by F-VHH-4, albeit with 300-fold lower efficiency. The presence of three consecutive serine residues in the CDR3 of F-VHH-4 could lead to more flexibility of this loop compared with that of F-VHH-L66 and allow F-VHH-4 to bind to an altered conformation of prefusion F. Although the possibility of selecting for an escape mutant could raise concerns for the clinical use of these VHHs, the escape virus was severely attenuated as it was impossible to grow the virus to substantial amounts to make high-titre virus stocks.

Other VHHs against RSV F have been described recently[19,35]. Nb017, for example, is a VHH developed by Ablynx that binds to antigenic site II of RSV F[19]. F-VHH-4 outperforms Nb017 by 15,000- and 180,000-fold on RSV A and RSV B neutralization respectively, reflecting the higher potency of prefusion-specific antibodies compared with antibodies that bind both conformations of F. To increase its neutralizing activity, Nb017 was re-formatted into a trivalent VHH named ALX-0171. However, monovalent F-VHH-4 and -L66 already display higher RSV A- and B- neutralizing activity than that reported for ALX-0171. In addition, the small size of monovalent VHHs may reduce their potential immunogenicity and facilitate penetration of the mucus layer that lines the respiratory epithelium to reach RSV-infected cells. It is also straightforward to link a monovalent VHH to other carriers, such as a VHH that binds to another epitope in RSV or to a protein with an effector function of interest.

A phase I/IIa clinical trial with inhaled ALX-0171 in hospitalized RSV-infected children indicated that the drug was safe. The trial also indicated that treatment with this trivalent pre- and postfusion F-reactive VHH could reduce viral replication and symptoms in the infected children. By selectively targeting prefusion F, for example with the VHHs described here, the antiviral impact in patients may be improved, which could result in a more robust clinical benefit in a treatment setting of RSV-infected patients. As the vast majority of RSV-infected infants are hospitalized at or after the peak of viral replication and disease, early RSV detection followed by rapid antiviral treatment remains key to reduce symptoms and prevent hospitalization of patients. Nebulization of F-VHH-4 as soon as possible after RSV confirmation offers the additional advantage that the antiviral is delivered directly to the site of infection.

Developing new antibody-based drugs for respiratory infections in a VHH format has a number of advantages. Production and purification are inexpensive, straight-forward and can be performed in organisms that are generally regarded as safe and without the use of animal-derived products. In addition, due to their small size, high stability and solubility, VHHs can be easily administered via inhalation, directly to the site of viral infection. The *in vitro* and *in vivo* results reported here support further testing of F-VHH-4 and F-VHH-L66 as novel therapeutics against RSV.

## Methods

**Study design.** It was our aim to develop efficient RSV-neutralizing VHHs that bind to the prefusion conformation of the F protein. Binding and neutralizing characteristics of the VHHs were analysed on HEK293T cells, Vero cells, HEp-2 or A549 cells. *In vivo* assays were performed in BALB/c mice. All animal experiments were conducted with the approval of the Ethical committee of the faculty of Sciences (University of Ghent, Belgium, EC number 2015-019).

**Generation of prefusion F-specific VHHs.** A llama was immunized subcutaneously six times with 167 μg DS-Cav1 each time at weekly intervals. The animal facility operates under the Flemish Government License Number LA1 700601. All experiments were done under conditions specified by law and authorized by the Institutional Ethical Committee on Experimental Animals (Ethical application 13-601-1).The DS-Cav1 protein was adjuvanted with poly (I:C) (Invivogen, 375 μg per injection) for the first two immunizations and with Gerbu LQ#3000 for the last four immunizations. One week after the last immunization 100 ml anticoagulated blood was collected for the preparation of peripheral blood lymphocytes. Total RNA from peripheral blood lymphocytes was used as template for first strand cDNA synthesis with oligodT primer. Using this cDNA, the VHH encoding sequences were amplified by a nested PCR using first the call 01 and call 02 primers (5′ GTCCTGGCTGCTCT TCTACAAGG3′, 5′GGTACGTGCTGTTGAACTGTTCC3′) and next the A6E and 38 primer (5′GATGTGCAGCTGCAGGAGTCTGGA/GGGAGG3′; 5′GGACTAGTGCGGCCGCTGGAGACGGTGACCTGGGT3′). The resulting PCR fragment was digested with PstI and NotI, and cloned in the PstI and NotI sites of the phagemid vector pHEN4. Electrocompetent *E. coli* TG1 cells were transformed with the recombinant pHEN4 vector. The resulting TG1 library stock was infected with VCS M13 helper phage to obtain a library of VHH-presenting phages.

One round of panning was performed on 20 μg of immobilized prefusion F (DS-Cav1) in one well of a microtiter plate (type II, F96 Maxisorp, Nunc). After blocking the prefusion F-coated well and an uncoated well used as a negative control with SEA BLOCK blocking buffer (Thermo Scientific), phages ($1 \times 10^{12}$ particles) were applied to these wells and incubated for 1 h at room temperature. After washing, the retained phage particles were eluted by applying a TEA-solution (14% triethylamine (Sigma) pH 10) for 10 min. Dissociated phages were transferred to a tube with 1 M Tris-HCl pH 8. Tenfold serial dilutions of the phages were used to infect TG1 cells, after which the bacteria were plated on LB agar plates with 100 μg ml$^{-1}$ ampicillin and 1% glucose.

**Detection of F-specific VHHs in bacterial periplasm.** After a single round of panning, ninety colonies were randomly selected for further analysis by ELISA for the presence of F-specific VHHs in their periplasm. The colonies were used to inoculate 2 ml of terrific broth (TB) medium with 100 μg ml$^{-1}$ ampicillin in 24-well deep well plates. After a 5 h incubation step at 37 °C, VHH expression was induced by adding isopropyl β-D-1-thiogalactopyranoside (IPTG) at a concentration of 1 mM. The plates were subsequently incubated overnight at 37 °C while shaking. The next day, bacterial cells were pelleted, resuspended in 200 μl TES buffer (0.2 M Tris-HCl pH 8, 0.5 mM EDTA, 0.5 M sucrose) and incubated at 4 °C for 30 min. Next, periplasmic extracts were prepared by adding water to induce osmotic shock. After 1 h incubation at 4 °C and subsequent centrifugation, the supernatants were collected. Microtiter plates were coated overnight with either 100 ng F protein in the postfusion conformation (FΔFP), 100 ng prefusion F (DS-Cav1) or bovine serum albumin (BSA, Sigma-Aldrich). The coated microtiter plates were blocked with 1% milk powder in phosphate-buffered saline (PBS) and 100 μl of the periplasmic extract was added to the wells. Bound VHHs were detected with anti-HA (1/2,000, MMS-101P Biolegend) mAb and horseradish peroxidase (HRP)-linked anti-mouse IgG (1/2,000, NXA931, GE Healthcare). After washing, 50 μl of TMB substrate (Tetramethylbenzidine, BD OptEIA) was added to every well. The reaction was stopped by addition of 50 μl of 1M $H_2SO_4$, after which the absorbance at 450 nM was measured with an iMark Microplate Absorbance Reader (Bio Rad). All periplasmic fractions for which the OD450 values obtained for prefusion or postfusion F were at least two times higher than the OD450 values obtained for BSA were selected and grown in 3 ml of LB medium with 100 μg ml$^{-1}$ ampicillin for plasmid isolation using the QIAprep Spin Miniprep kit (Qiagen). The cDNA sequence of the VHH was determined by Sanger sequencing using the M13RS primer (5′CAGGAAACAGCTATGACC3′).

**Expression of VHHs in Pichia pastoris.** To clone VHH sequences into a yeast expression vector, the VHH coding sequences were PCR amplified from the respective pHEN4 plasmids using the following forward and reverse primers (5′GGCGGGTATCTCTCGAGAAAAGGCAGGTGCAGCTGCAGGAGTCT GGG3′; 5′CTAACTAGTCTAGTGATGGTGATGGTGGTGGCTGGAGACG GTGACCTGG3′). The resulting PCR products were digested with XhoI and SpeI and ligated into XhoI/SpeI-digested pKai61 backbone. The origin of the pKai61 vector is described in Schoonooghe et al.[36]. The VHH sequences are cloned in frame with a slightly modified version of the *S. cerevisiae* a-mating factor signal sequence. The encoded genes contain a C-terminal 6XHis tag and are under control of the methanol inducible AOX1 promoter. The plasmid

contains a Zeocine resistance marker for selection in bacterial as well as in yeast cells. The vectors were linearized by PmeI before transformation of *P. pastoris* strain GS115 using the condensed transformation protocol described by Lin-Cereghino *et al.*[37].

**Purification of VHHs produced by Pichia pastoris.** Expression of VHH by transformed *P. pastoris* clones was first analysed in 2 ml cultures. On day one, individual transformants were used to inoculate 2 ml of YPNG medium (2% pepton, 1% Bacto yeast extract, 1.34% YNB, 0.1 M potassium phosphate pH 6, 0.00004% biotine, 1% glycerol) with 100 µg ml$^{-1}$ Zeocin (Life Technologies) and incubated while shaking at 28 °C for 24 h. Next, cells were pelleted and the YPNG medium was replaced by YPNM medium (2% peptone, 1% Bacto yeast extract, 1.34% YNB, 0.1 M potassium phosphate pH 6, 0.00004% biotin, 1% methanol) to induce VHH expression. Cultures were incubated at 28 °C while shaking for 48 h. Fifty microliters of 50% methanol was added to the cultures at 16, 24 and 40 h. After 48 h, the yeast cells were pelleted and the supernatant was retained to assess the presence of VHH. To select VHHs with RSV-neutralizing activity, serial dilutions of the crude YPNM supernatant from individual *P. pastoris* transformants were tested in a plaque-reduction assay.

*P. pastoris* transformants that yielded high levels of VHH in the medium and with high RSV-neutralizing activity were selected for scaling up using 100 or 300 ml *P. pastoris* cultures. Growth and methanol induction conditions and harvesting of medium were similar as mentioned above for the 2 ml cultures. The cleared medium was subjected to ammonium sulfate precipitation (80% saturation) for 4 h at 4 °C. The insoluble fraction was pelleted by centrifugation at 20,000g and solubilized in 10 ml HisTrap binding buffer (20 mM sodium phosphate, 0.5 M NaCl, 20 mM imidazole, pH 7.4) before purification on a 1 ml HisTrap HP column (GE Healthcare). Relevant fractions containing the VHH were pooled, and concentrated with a Vivaspin column (5 kDa cutoff, GE Healthcare) and then subjected to size-exclusion chromatography (Superdex 75). Fractions containing the VHH were again pooled and concentrated and the concentration was determined. The purified VHHs were aliquoted and stored at − 80 °C until further use.

**Cells and viruses.** HEp-2 cells (ATCC, CCL-23), Vero cells (ATCC, CCL-81), HEK-293 T cells (a gift from Dr M. Hall, University of Birmingham, Birmingham, UK) and A549 cells (ATCC, CCL-185) were grown in Dulbecco's modified eagle medium (DMEM) supplemented with 10% heat-inactivated fetal calf serum (FCS), 2 mM L-glutamine, non-essential amino acids (Invitrogen, Carlsbad, CA, USA) and 1 mM sodium pyruvate at 37 °C in the presence of 5% carbon dioxide. RSV A2, an A subtype of RSV (ATCC, VR-1540, Rockville), RSV B49, a B subtype of RSV (BE/5649/08 clinical strain[38], source described in ref. 38, obtained from Prof Marc Van Ranst, KU Leuven, Leuven, Belgium), RSV A Long (ATCC, VR-26, kind gift from Dr Rik De Swart, Erasmus MC, Rotterdam, The Netherlands), MAD/GM2_2/12, MAD/GM2_12/12, MAD/GM2_13/12, MAD/GM2_14/12, MAD/GM3_10/14, MON/9/92 (primary RSV A strains, isolated in Madrid (MAD) or Montevideo (MON)) and MAD/GM3_7/13 (a primary RSV B strain isolated in Madrid) were propagated in HEp-2 cells and quantified on Vero cells by plaque assay using goat anti-RSV serum (AB1128, Chemicon International). Clinical isolates MAD/GM2_2/12, MAD/GM2_12/12, MAD/GM2_13/12, MAD/GM2_14/12, MAD/GM3_10/14 and MON/9/92 represent at least 2 different genotypes of antigenic group A[39,40].

**Plaque-reduction assay.** A dilution series of the VHHs/mAbs was prepared in Opti-MEM (Gibco), incubated with RSV for 30 min at 37 °C and used to infect confluent Vero cells. After 3 h, an equal volume of 1.2% avicel RC-851 Q(FMC Biopolymers) in DMEM medium supplemented with 2% FCS, 2 mM L-glutamine, non-essential amino acids and 1 mM sodium pyruvate was added to each well and the infection was allowed to continue at 37 °C for three days. Viral infection was tested by immunostaining of the viral plaques with goat anti-RSV serum (AB1128, Chemicon International) and horseradish peroxidase-conjugated anti-goat IgG (SC2020, Santa Cruz). The plaques were visualized by applying TrueBlue peroxidase substrate (KPL, Gaithersburg). For the post-attachment neutralization assay, a dilution series of VHH/mAb was added to HEp-2 cells that had been preincubated with RSV at 4 °C for 2 h. Control antibody 65 is a mouse IgG2a monoclonal antibody specific for the influenza matrix protein 2 (ref. 41).

**Recombinant mKate-RSV panel.** The pSynkRSV-line19F, a BAC system containing RSV cDNA was used to generate recombinant RSV antigenomes with matched G and F genes from various RSV strains to recover new infectious clones[42]. The panel of reconstructed recombinant mKate-RSV consists of five RSV subtype A and three RSV subtype B viruses, which represent RSV lab strains and primary isolates from distinct temporal and geographic regions. The RSV subtype A panel comprises A2/D46 and A2/L19, which are laboratory strains, and three primary strains: Riyadh 91/2009, 2-20 F/G and A1998/12-21. RSV subtype B panel comprises one lab strain B/18537 and two primary isolates TX11-56 and NH1276.

**Fluorescence plate reader neutralization assay.** Fluorescence plate reader neutralization assay was performed as previously described[13]. A total of 2.4 × 10⁴ HEp-2 cells per well in 30 µl culture medium were seeded in 384-well black optical bottom plate (Nunc 384-well plates, Thermo Scientific). Serum samples or antibodies (1 mg ml$^{-1}$) were diluted four-fold for 12 dilutions starting from 1:10, equal volume of recombinant mKate-RSV (subtype A and subtype B) was added and mixed, incubated at 37 °C for 1 h, then 50 µl mixture of sample and virus was added to cells and incubated at 37 °C for 22–24 h. After incubation, assay plate was measured for fluorescence intensity in microplate reader at Ex 588 nm and Em 635 nm (SpectraMax Paradigm, molecular devices). For the 96-well plate format, 5 × 10⁴ cells per well in 100 µl of culture medium were seeded in a 96-well plate, 100 µl mixtures of sample and virus were added to the plate. Virus inhibition was measured as the reduction of fluorescence of samples comparing to virus control. Data was analysed and EC$_{50}$ was calculated by curve fitting with GraphPad Prism software (GraphPad Software).

**Attachment assay.** RSV A2 (1 × 10⁷ PFU) was incubated with 1 µM VHH/antibody or 2.5 µM dextran sulfate (MP Biomedicals) for 30 min at room temperature. The mixtures were added to chilled HEp-2 cells and incubated for 2 h at 4 °C. The cells were washed five times with cold PBS with 0.5% BSA (PBS-BSA). Half of the cells were fixed with 2% PFA, stained with goat anti-RSV polyclonal serum (AB1128, Chemicon International) and subsequently with AlexaFluor 488 labelled donkey anti-goat antibody (Invitrogen, 1/600 dilution). After washing, the cells were analysed using a FACSCalibur flow cytometer. The other half of the cells was plated out and incubated at 37 °C in DMEM with 2% FCS. After 48 h, the cells were collected, fixed and analysed as described above.

**VHH binding to cells expressing F.** HEK293T cells were transfected with pCAGGS-F, which encodes a codon-optimized RSV F cDNA, with the FuGENE HD transfection reagent (Promega) according to the manufacturer's protocol. To trace transfected cells, transfections were performed in the presence of peGFP-NLS. Control transfections were performed with only peGFP-NLS. Forty-two hours after transfection the cells were detached, washed and blocked. Subsequently, the cells were incubated with a 1/10 dilution series of VHH or mAb in PBS-BSA. One hour later the cells were washed and stained with mouse anti-Histidine Tag antibody (MCA1396, Abd Serotec) followed by anti-mouse IgG Alexa 633 (Invitrogen) (for the VHH samples) or anti-human IgG Alexa 633 (Invitrogen) (for the antibody samples). The stained cells were analysed using a FACSCalibur flow cytometer. All procedures were performed on ice or at 4 °C.

Binding to infected cells was assessed in a similar manner. Vero cells were inoculated with RSV A2 at a MOI of 0.1 and after 48 h the cells were collected and fixed with 2% PFA. The cells were stained with 1 µg ml$^{-1}$ of VHH or antibody. The rest of the staining procedure was performed as described above.

**Immunostaining of infected cells.** A549 cells, grown on glass plates for confocal imaging or µ-Dish glass bottom dishes (ibidi) for TIRF-imaging, were mock-infected or infected with RSV A2 (MOI 1) for 24 h and fixed with 2% PFA. The cells were blocked and stained with 1 µg ml$^{-1}$ of F-VHH-4, F-VHH-L66 or Ctrl-VHH in PBS-BSA. One hour later the cells were washed and fixed again with 2% PFA and stained with polyclonal goat anti-RSV serum (or polyclonal rabbit anti-G serum (Sino Biological Inc. 11070-V08H) for the TIRF images) and mouse anti-Histidine Tag antibody. VHH binding was detected with anti-mouse IgG Alexa 488 and anti-RSV serum with anti-goat Alexa 633 (or anti-rabbit Alexa 568 for the TIRF images). After washing the samples were mounted and confocal images were recorded with a SP5 Leica confocal microscope. TIRF images were recorded with a Zeiss TIRF Observer. The recorded images were processed with Volocity software (Perkin Elmer).

**Cross-competition analysis with biolayer interferometry.** Antibody cross-competition was performed as described previously[11] with some modifications. Briefly, DS-Cav1 protein (10 µg ml$^{-1}$) was immobilized in a random orientation on AR2G biosensors (ForteBio) pre-activated with N-hydroxysuccinimide (NHS) and 1-ethyl-3-(3-dimethylaminopropyl)carbodiimide (EDC) through amine coupling reaction in acetate buffer (pH 5). The reaction was quenched by 1 M ethanolamine (pH 8) and DS-Cav1-immobilized biosensors were then equilibrated with assay buffer (PBS with 1% BSA). The biosensors were dipped in competitor antibodies (35 µg ml$^{-1}$ in assay buffer) for 300 s followed by analyte antibodies (35 µg ml$^{-1}$ in assay buffer) for 300 s with a short baseline step (60 s) in between the two antibody steps. All assays were performed at a set temperature of 30 °C with agitation of 1,000 r.p.m. in an Octet HTX instrument (ForteBio). Per cent inhibition of antibody binding by competing mAbs was calculated with the following equation: Inhibition (%) = 100 − [(analyte antibody binding in the presence competitor mAb)/(analyte antibody binding in the presence of isotype control mAb)] × 100.

**F-VHH escape mutant viruses.** A dilution series of RSV A Long was used to infect Vero cells in the presence of a dose corresponding to 20, 80 or 320 times the $IC_{50}$ of F-VHH-4 or F-VHH-L66. After 6 days of infection, the supernatant of the cells infected with the highest dilution of RSV for which virus could be detected (observed by plaque assay), was used to infect a subsequent series of Vero cells in the presence of VHH or palivizumab. After two and four passages in presence of F-VHH-L66 and F-VHH-4, respectively, single RSV isolates were obtained by serial dilution which were amplified on Vero cells. Total RNA was isolated from the resulting virus stocks, cDNA was prepared and the F gene was amplified and sequenced with following FW and RV primers (5′ATCAAGCTTT AACAATGGAGTTGCTAATCCTCA3′, 5′ AACCGCTCGAGTTTAGTTACT AAATGCAATAT3′). A plaque-reduction assay with these virus isolates was performed.

**Surface plasmon resonance.** Purified DS-Cav1 with a C-terminal Strep-tag II and 6X HisTag was captured on an NTA sensor chip to ∼530 response units (RU) each cycle using a Biacore X100 (GE Healthcare). The NTA sensor chip was regenerated between cycles using 0.25 M EDTA followed by 0.5 mM $NiCl_2$. A buffer-only sample was injected over the DS-Cav1 and reference flow cells, followed by F-VHH-4 or F-VHH-L66 2-fold serially diluted from 5 nM to 39.1 pM in HBS-P +, with a duplication of the 1.25 nM concentration. The data were double-reference subtracted and fit to a 1:1 binding model using Scrubber.

**Protein production for crystallization.** Freestyle 293-F cells (R79007, Invitrogen) were transfected with plasmid encoding stabilized prefusion RSV F (DS-Cav1)[13]. Proteins were expressed in the presence of kifunensine (5 μM) and purified from cell supernatants using Strep-Tactin resin (IBA). Tags were removed by thrombin digestion and glycans were removed by digestion with Endo H (10% w/w) for 2 h at room temperature. Purified DS-Cav1 was combined with a 1.5-fold and 2-fold molar excess of *Pichia*-expressed F-VHH-4 and F-VHH-L66, respectively, and separated from excess VHH using a Superose6 column (GE Healthcare Biosciences). For crystallization trials of the unbound VHHs, vectors encoding F-VHH-4 or -L66 with a C-terminal 3C cleavage site, Twin-Strep-tag and 8X HisTag were transfected into FreeStyle 293-F cells. Proteins were purified from cell supernatants using Strep-Tactin resin before tag removal by HRV 3C digestion. VHHs were separated from cleaved tags using a Superdex 75 column.

**Crystallization and data collection.** F-VHH-4 crystals were produced by sitting-drop vapour diffusion by mixing 100 nl of F-VHH-4 (9.45 mg ml$^{-1}$) with 50 nl of reservoir solution containing 3% (v/v) PEG 400, 2.2 M ammonium sulfate, 0.2 M sodium chloride and 0.1 M HEPES pH 7.5. Crystals were soaked in reservoir solution lacking 0.2 M sodium chloride and supplemented with 25% 2R,3R-butanediol and frozen in liquid nitrogen. Diffraction data were collected to 1.87 Å at SBC beamline 19-ID (Advanced Photon Source, Argonne National Laboratory).

Crystals of F-VHH-4 in complex with DS-Cav1 were produced by hanging-drop vapour diffusion by mixing 2 μl of the complex (2.45 mg ml$^{-1}$) with 1 μl of reservoir solution containing 0.2 M ammonium citrate pH 4.5, 14.75% (w/v) PEG 3350 and 8.85% (v/v) isopropanol. Crystals were soaked in reservoir solution supplemented with 30% (v/v) ethylene glycol and frozen in liquid nitrogen. Diffraction data were collected to 3.3 Å at the GM/CA beamline 23-IDB (APS, Argonne National Laboratory).

Crystals of F-VHH-L66 in complex with DS-Cav1 were produced by free-interface diffusion by mixing 2 μl of the complex (4.79 mg ml$^{-1}$) with 2 μl of 0.05 M potassium phosphate and 20% (w/v) PEG 4000 in a Crystal Former Optimization Chip (Microlytic). Crystals were soaked in crystallization solution supplemented with 30% (v/v) ethylene glycol and frozen in liquid nitrogen. Diffraction data were collected to 3.8 Å at SBC beamline 19-ID.

**Structure determination.** All data were indexed and integrated in iMOSFLM[43] and scaled and merged using AIMLESS[44]. F-VHH-4 crystals formed in space group $P2_12_12_1$ with near-perfect pseudo-merohedral twinning to appear as space group $P4_12_12$. Diffraction data from F-VHH-4 crystals were initially processed as $P422$ and a molecular replacement solution containing one VHH per asymmetric unit in space group $P4_12_12$ was obtained using PHASER[45]. However, refinement resulted in suspiciously high $R_{work}/R_{free}$ values, particularly given the high resolution of this data set. Evaluation of systematic absences revealed that 00 $l$ reflections were present for $l = 2n$, which is compatible with a $4_2$ or $2_1$ screw axis, but not with the $4_1$ axis found in the molecular replacement solution. Evaluation of the L- and H-tests revealed that the crystal was pseudo-merohedrally twinned (Supplementary Fig. 17a) with a twin fraction greater than 0.4 (Supplementary Fig. 17b). The data were subsequently reprocessed in space group $P2_12_12_1$ and a molecular replacement solution was obtained containing two VHH molecules per asymmetric unit. After manual model building in Coot[46], the structure was refined in PHENIX[47] to an $R_{work}/R_{free}$ of 17.0/20.8% using twin law $kh$-$l$ and a refined twin fraction of 0.47.

The F-VHH-4 and -L66 complexes formed crystals in space groups $P3_221$ and $P3_121$, respectively. The unbound F-VHH-4 structure and the previously solved prefusion F structure (PDB ID: 4MMS) were used as search models for molecular replacement. The F-VHH-4 and -L66 structures were built manually in

Coot and refined with PHENIX using NCS torsion restraints and reference-model restraints to an $R_{work}/R_{free}$ of 18.4/23.4% and 24.8/28.6%, respectively. Complete data collection and refinement statistics are presented in Table 1 and stereo views of a portion of the electron density for each structure are shown in Supplementary Fig. 18.

**HMPV microneutralization test.** Predetermined amounts of GFP-expressing hMPV recombinant viruses (NL/1/00 A1 sublineage or NL/1/99 B1 sublineage, a kind gift of Bernadette van den Hoogen and Ron Fouchier, Rotterdam, the Netherlands) or GFP-hRSV (A2 strain, a kind gift of Mark Peeples, Columbus, Ohio, USA) were mixed with serial dilutions of VHH or monoclonal antibodies before being added to cultures of Vero-118 cells. Forty-eight hours later, the medium was removed, PBS was added and the amount of GFP per well was measured with a Tecan microplate reader M200. Fluorescence values were represented as per cent of a virus control without antibody and $IC_{50}$ values were calculated from the plotted curves.

**Mice.** Specific pathogen-free, female BALB/c mice were obtained from Charles River (Charles River Wiga, Sulzfeld, Germany). The animals were housed in a temperature-controlled environment with 12 h light/dark cycles; food and water were provided ad libitum. The animal facility operates under the Flemish Government License Number LA1400536. All experiments were done under conditions specified by law and authorized by the Institutional Ethical Committee on Experimental Animals (Ethical application EC2015-019).

**Administration of VHHs and RSV A2 infection in mice.** Mice, seven weeks old, were randomly distributed in experimental groups of six animals. They were slightly anesthetized by isoflurane before intranasal administration of VHH, palivizumab or RSV challenge virus. VHH, palivizumab and RSV virus were administered intranasally in a total volume of 50 μl PBS. Each group received 10 μg of F-VHH-4, 10 μg of Ctrl-VHH or 10 μg of palivizumab 4 h before infection with $10^6$ PFU of RSV A2. The mock-infected group received 10 μg of Ctrl-VHH that is specific for a Junin virus protein.

**Determination of lung viral titres by plaque assay.** Five days after challenge, mice were euthanized, lungs were removed and homogenized by vigorous shaking with a Mixer Mill MM 2000 (Retsch) in the presence of a sterile metal bead in 1 ml HBSS containing 20% sucrose and supplemented with 1% penicillin and 1% streptomycin. Lung homogenates were cleared by centrifugation at 4 °C and used to titrate the virus by plaque assay. We set lung homogenates in which no virus was detected as the detection limit of the assay.

**Determination of lung viral titre by qRT-PCR.** To determine the lung RSV load by qRT-PCR, total RNA from the cleared lung homogenates was prepared by using the High Pure RNA Tissue Kit (Roche) according to the manufacturer's instructions. Next, cDNA was prepared by the use of random hexamer primers and the Transcriptor First Strand cDNA Synthesis Kit (Roche). The relative levels of genomic RSV M cDNA were determined by qRT-PCR using primers specific for the RSV A2 M gene (5′TCACGAAGGCTCCACATACA3′ and 5′GCAGGG TCATCGTCTTTTTC3′) and a nucleotide probe (#150 Universal Probe Library, Roche) labelled with fluorescein (FAM) at the 5′-end and with a dark quencher dye near the 3′-end. The qRT-PCR data were normalized to mRPL13A mRNA levels.

**Analysis of pulmonary cell infiltration.** The immune cell composition of bronchoalveolar lavages (BAL) was determined by analysing surface expression of MHC-II, CD3e, SiglecF, CD4, CD8, CD11b and CD11c. Briefly, high-affinity Fc receptors (FcRs) were blocked by incubation with purified anti-mouse CD16/CD32 (Fc Block, BD Pharmingen, 553142) for 30 min at 4 °C. Subsequently, the cells were stained with MHC-II-eFluor450 (eBioscience, 48-5321-82, 1/500), CD3e-AlexaFluor488 (BD Pharmingen, 557666, 1/300), SiglecF-PE (BD Pharmingen, 562068, 1/200), CD4-PerCP (BD Pharmingen, 553052, 1/300), CD8-PE-Cy7 (eBioscience, 25-0081-81, 1/300), CD11b-APC-Cy7 (BD Pharmingen, 557657, 1/500) and CD11c-APC (BD Pharmingen, 550261, 1/200) for 1 h at 4 °C. After staining, samples were measured on an LSR II flow cytometer (BD Biosciences, San Jose, CA, USA), and analysed using FlowJo X (TreeStar) software.

**Testing lower doses of F-VHH-4 in mice.** BALB/c mice, 7-weeks-old, were slightly anesthetized by isoflurane before intranasal administration of VHH or RSV challenge virus. VHH and RSV virus were formulated in PBS and administrated in a total volume of 50 μl, which was distributed equally over the two nostrils. Each group of six mice received either 1, 3 or 10 μg of F-VHH-4 or 10 μg of Ctrl-VHH 4 h before infection with $10^6$ PFU of RSV A2. The mock-infected group received 10 μg of Ctrl-VHH.

**Statistical analysis.** We used Graphpad Prism 6 for statistical analyses, and the two-sided Mann–Whitney U test to evaluate differences between two groups.

**Data availability.** Coordinates and structure factors for the unbound F-VHH-4, and the F-VHH-4 and -L66 bound prefusion F complex structures have been deposited in the Protein Data Bank under accession codes 5TP3, 5TOJ, and 5TOK, respectively. All other relevant data are available from the authors upon request.

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

## Acknowledgements

We thank Amanda Gonçalves and Eef Partheons from VIB Bio Imaging Core, Liesbeth Vande Ginste, Lien Van Hoecke, Soraya Van Cauwenberghe and Emilie Shipman for excellent technical assistance and Dr Florencia Linéro for providing the control VHH. This study was supported by IWT-Vlaanderen (Ph.D. student fellowship to I.R.), National Institute of General Medical Sciences of the National Institutes of Health award T32GM008704 (M.S.A.G.) and P20GM113132 (J.S.M.), FWO-Vlaanderen (Postdoctoral fellowship to B.S.), Ghent University Special Research Grant BOF12/GOA/ 014, Interuniversity Attraction Poles programme (IAP7, BELVIR), VIB and Mineco (Spain) Grant SAF2015-67033-R (J.A.M.). Results shown in this report are derived in part from work performed at Argonne National Laboratory, Structural Biology Center at the Advanced Photon Source. Argonne is operated by UChicago Argonne, LLC, for the U.S. Department of Energy, Office of Biological and Environmental Research under contract DE-AC02-06CH11357. Data in this report were also obtained at GM/CA@APS,

which has been funded in whole or in part with Federal funds from the National Cancer Institute (ACB-12002) and the National Institute of General Medical Sciences (AGM-12006).

## Author contributions

I.R., J.S.M., B.S.G., B.S. and X.S. planned the study. I.R., M.S.A.G., S.C.K., K.S., D.W., M.K., M.C., V.M. and J.S. performed the research. I.R., M.S.A.G., B.S., J.S.M. and X.S. wrote the manuscript with contributions from J.A.M. and B.S.G. All authors reviewed the manuscript before submission.

## Additional information

**Competing financial interests**: X.S., B.S., I.R., J.S.M., M.S.A.G. and B.S.G. are named as inventors on a patent pending entitled: 'Single-domain antibody against RSV F protein', US 62/181,522. The remaining authors declare no competing financial interests.

DOI: 10.1038/ncomms16165　　**OPEN**

# Corrigendum: Potent single-domain antibodies that arrest respiratory syncytial virus fusion protein in its prefusion state

Iebe Rossey, Morgan S.A. Gilman, Stephanie C. Kabeche, Koen Sedeyn, Daniel Wrapp, Masaru Kanekiyo, Man Chen, Vicente Mas, Jan Spitaels, José A. Melero, Barney S. Graham, Bert Schepens, Jason S. McLellan & Xavier Saelens

*Nature Communications* 8:14158 doi: 10.1038/ncomms14158 (2017); Published 13 Feb 2017; Updated 29 Nov 2017.

In this Article, the clinical isolate MAD/GM3_10/14 of respiratory syncytial virus (RSV) is incorrectly described as an RSV A isolate. MAD/GM3_10/14 has been identified as an RSV B isolate on the basis of its fusion protein sequence. The fusion protein sequences of the clinical isolates used in this study have been deposited in GenBank under accession codes MF361899 (MAD/GM2_2/12), MF361900 (MAD/GM2_12/12), MF361901 (MAD/GM2_13/12), MF361902 (MAD/GM2_14/12), MF361903 (MAD/GM3_10/14), MF361904 (MON/9/92), and MF361905 (MAD/GM3_7/13).

