## [Peer Review File · Nature Communications]

Reviewers' comments:

Reviewer #1 (Remarks to the Author):

The authors describe the isolation and analysis of two llama single-domain antibodies that bind tightly to prefusion FSV F protein. The data presented are extensive with three crystal structures, cell-based assays, affinity data and experiments with mice. RSV is a major pathogen for both children and the elderly, and there are currently no anti-RSV vaccines and anti-viral medications. Thus, these results are of high interest, as the molecules described may be used as possible therapeutics. The paper is very exciting and well written and I recommend publication with minor modifications outlined below.

Specific comments:

p. 4, heavy-chain only antibodies are not only found in camelids, but also exist in sharks and their relatives

p. 6, The first mention of the DS-Cav construct is in the Results section, and readers not familiar with work from these authors may not know what it is. A brief explanation of the properties and modifications introduced into this construct would be useful in the introduction (where you briefly mention that such protein engineering has been done), especially as this construct was possibly a key to the discovery of these two nanobodies.

p. 7, Please give the Kabat residue numbers for the cysteine residues in the VHH4 and L66 CDR2 and CDR3 residues.

p. 17, for the EndoH digestion, how long and at what temperature was the digestion carried out?

p. 18, please provide in the supplemental materials an L-plot for the twinned crystals, and somewhere report the twin fraction. The comment about determining the twinning based on the systematic absences is not clear, nor proof enough to declare that a crystal is twinned. Also, twinning of P212121 to appear tetrahedral is 'pseudo-merohedral' twinning, not 'merohedral' twinning (see Int. Tables, Vol. C. p. 12, Tables 1.3.4.2 and 1.3.4.1. and also a good paper describing a similar twinning problem that may be helpful (see Acta Cryst. (2010). D66, 163–175).

p. 18, Only the first letter in 'Coot' should be in caps, 'PHENIX' should be in all caps- these appear in multiple formats in the methods section.

In Supplementary Figure 2, the wells that are boxed look like the other wells to me- are we supposed to see something here?

In Supp. Figure 6, should there be standard deviation markers on the bars?

In regards to Supp. Figure 8, here or in the text, please report the RMSD between the bound and unbound VHH-4 structures.

While VHH-4 loses some affinity binding monomeric F as compared to trimeric F, VHH-L66 does not bind the monomeric protein at all. Can the authors speculate as to the differences in affinity for the monomer, based on the crystal structures?

In Supp. Table 2, for the 1.87A structure, more reflections were used in refinement than were collected, and for the other two structures the numbers of reflections also differ by quite a lot- please check these numbers carefully. Also there should be a comment somewhere in the text or table about the identity of the ligand/ion for the two low-resolution structures.

Reviewer #2 (Remarks to the Author):

The manuscript by Saelens et al describes the generation and isolation of a single domain antibody from llama that is highly potent in its ability to neutralize RSV A and B strains. The authors map the epitope of its binding on the F protein using escape mutation and X-ray crystallography, and show the epitope to be present on pre-F form of the fusion protein and binding to two adjacent protomers. Furthermore they demonstrate in vivo efficacy of the nanobody via delivering it intra-nasally into mice followed by RSV challenge. Overall based on their results the author's claim that their antibody could be used in humans to prevent RSV infection prophylactically.

The manuscript could benefit from clearly laid out objective discussion as suggested below;

- 1) The in vitro potency of the antibody is derived from the plaque assay and is very much dependent on the input PFU. Many scientists in RSV research use 50-100 pfu as input virus for their experiments. However the current manuscript utilizes as low as 25-30 pfu only which could lead to lower or "ultra-potent" IC50's. In defense of the authors they do have comparator antibodies in the same assay and their nanobody is an order more potent than D25. It would be important to include the input virus pfu upfront in the figure and discuss the caveat of very low potency explained in parts due to use of lower viral pfu in their assays for neutralizations.

- 2) The table in Figure 2b does not match well with the figure 5A published earlier by Ngwuta and Graham et al (Sci Transl Med. 2015 October 14; 7(309): 309ra162.

doi: 10.1126/scitranslmed.aac4241.). Specifically In earlier manuscript 101F seems to compete with MPE8 (60-80% competition) but in the current manuscript it is much lower and more discrepant between the two modes of competition set up(4% and 40%). Furthermore, competition between 101F and AM14 is about 57% or 91% depending on the how the competition is done, while in the 2015 manuscript there was no difference – 85% both ways. Since Dr. Graham is also an author on the current manuscript the authors should be able to reconcile the differences and address them in the manuscript.

- 3) The authors suggest that the nanobody could be developed as a therapeutic for humans. It has been established using several antibodies (Palivizumab, Motavizumab, D25/MEDI8897 and REGN2222) that the cotton rat model is more useful in translating animal studies and amounts of antibody required for efficacy in humans. However, the authors chose to use the mouse model for their in vivo studies. Therefore it would be interesting to see the in vivo experiments be conducted in cotton rats to ascertain human clinical doses and comparison

to ALX-0171

4) Another minor point from a clinical development potential of VHH antibody is the disulfide bond found in CDR's which could turn out to be a manufacturing liability

5) The most interesting data are on how the antibody binds to the pre-fusion F of RSV. These studies reveal that the antibody binds in a unique manner so far not seen with the human antibodies isolated and reported. The antibody binds to two adjacent protomer in a cavity with Thr50 in the pit of the cavity. This binding is similar to what has been reported for MPE8 (a siteIII binding human antibody) yet different that it engages additional surrounding residues not conserved with hMPV F protein (and therefore cannot neutralize hMPV). Furthermore it competes with Site II (Palivizumab), SiteIV (101F) and SiteV (AM14) binding antibodies. One interesting possibility is that due to the nanobody nature of VHH it may be able to access the cavity at the interface of two protomers and hence no human antibody (due to the size of the IgG format) is found that bind in this unique fashion. It is important to note that some site III human antibodies are more potent in neutralizing RSV as Fab than as full length antibodies (unpublished data). Therefore it would be of interest for the readers to see data of a monovalent Fab of MPE8 compared to full length MPE8 and VHH in an in vitro neutralization assay.

6) The readers could also benefit from a broader survey of neutralization potency on actual clinical isolates of both RSV A and B strains.

Reviewer #3 (Remarks to the Author):

This manuscript describes the generation of llama single-domain monoclonal antibodies specific for respiratory syncytial virus pre-fusion F protein and their characterization. The data presented include detailed presentation of the methods of generation of the antibodies as well as detailed structural studies of the antibodies alone or in a complex with the pre-fusion F protein and analyses of the biological activities of the antibodies both in tissue culture and in animals.

The manuscript is clearly written, very interesting, and describes development of two novel antibodies that define a new antigenic site on the pre-fusion F protein, albeit a site that is not likely accessed by human antibodies as indicated by the authors' structural analysis. The results of in vitro and animal studies strongly support the authors' proposal that these antibodies have significant potential as prophylaxis or possible treatment of RSV induced disease. These antibodies also have significant potential as reagents for basic studies of virus infection, including virus entry and refolding of the pre-fusion F to the post fusion form. Indeed their analysis indicates that inhibition of infection by these antibodies may be due to stabilization of the trimer structure of the pre-fusion F protein. In general, the data are very comprehensive and the quality of results is mostly excellent.

There are several novel aspects of the study. First, the authors have defined a new antigenic site accessed only by single domain antibodies. Second, they describe a novel potential prophylactic or therapeutic agent unlike any previously tested for RSV. Third, they describe a new reagent for basic studies of virus infection.

The authors do describe the generation of the antibodies in detail. It is not clear that such detail is necessary since protocols are moderately widely available and offered by numerous companies.

There are some minor problems that should be addressed.

1. Page 3, line 20: a reference for antibody preventing F remodeling needs to be added.
2. Page 4, line 11: references for the use of VHHs antibodies for treatment of rheumatoid arthritis and cancer should be added.
3. Page 10, lines 5-8, supplementary fig 10: a description of the generation of monomer F protein needs to be added to materials and methods or at least a reference for preparation of this form should be added.
4. Page 11, line 13: text indicates lungs in Supplemental Fig 14 but in the figure the data are described as characterizing BAL. Which is it?
5. Some of the Y-axes on figures are not clearly labeled: Figure 1a, 1b, 1d, 1f; Figure 3 both panels; Figure 6a, 6b, and 6c; Supplemental figure 4: Y-axis needs a label. Label on the side of the chart in Fig 2b is unclear.
6. Supplemental figure 7: This is the only figure that is of questionable quality. The signals in panels in A, particularly the red signal, are very hard to see. This figure could be improved.

Reviewer #1 (Remarks to the Author):

The authors describe the isolation and analysis of two llama single-domain antibodies that bind tightly to prefusion FSV F protein. The data presented are extensive with three crystal structures, cell-based assays, affinity data and experiments with mice. RSV is a major pathogen for both children and the elderly, and there are currently no anti-RSV vaccines and anti-viral medications. Thus, these results are of high interest, as the molecules described may be used as possible therapeutics. The paper is very exciting and well written and I recommend publication with minor modifications outlined below.

Author response: We thank the reviewer for this appreciation of our work.

Specific comments:

p. 4, heavy-chain only antibodies are not only found in camelids, but also exist in sharks and their relatives

Author response: On page 4 we have included sharks as an example of a species with heavy-chain only antibodies and for that refer to Greenberg et al., Nature 1995.

p. 6, The first mention of the DS-Cav construct is in the Results section, and readers not familiar with work from these authors may not know what it is. A brief explanation of the properties and modifications introduced into this construct would be useful in the introduction (where you briefly mention that such protein engineering has been done), especially as this construct was possibly a key to the discovery of these two nanobodies.

Author response: We agree that including a description of the prefusion F protein used to isolate these nanobodies will be helpful for the broad readership of *Nature Communications* and have therefore modified the introduction to include the following (lines 54-60):

“However, intensive screening for human mAbs that potently neutralize RSV has resulted in the isolation of prefusion F-specific antibodies with more robust neutralizing activity than palivizumab. Recently, RSV F was successfully stabilized in its prefusion conformation through the introduction of an additional intraprotomeric disulfide bond, cavity-filling mutations, and a trimerization motif. This reagent, called DS-Cav1, has been instrumental in revealing that the vast majority of RSV-neutralizing immunoglobulins in human sera selectively bind to F in its prefusion conformation.”

p. 7, Please give the Kabat residue numbers for the cysteine residues in the VHH4 and L66 CDR2 and CDR3 residues.

Author response: We have now used Kabat residue numbering for the VHHs. We have modified the text to read:

“F-VHH-4 and -L66 both have a cysteine residue at position 100c in the CDR3 and at position 50 in the CDR2.” Lines 112-113

“Predicted amino acid sequences of F-VHH-4 and F-VHH-L66 with Kabat numbering” Legend of Figure 1, panel c.

p. 17, for the EndoH digestion, how long and at what temperature was the digestion carried out?

Author response: We have added this information to lines 359-360 of the Methods section, which now reads:

“Tags were removed by thrombin digestion and glycans were removed by digestion with Endo H (10% w/w) for 2 hours at room temperature.”

p. 18, please provide in the supplemental materials an L-plot for the twinned crystals, and somewhere report the twin fraction. The comment about determining the twinning based on the systematic absences is not clear, nor proof enough to declare that a crystal is twinned. Also, twinning of $P2_12_12_1$ to appear tetrahedral is ‘pseudo-merohedral’ twinning, not ‘merohedral’ twinning (see Int. Tables, Vol. C. p. 12, Tables 1.3.4.2 and 1.3.4.1. and also a good paper describing a similar twinning problem that may be helpful (see Acta Cryst. (2010). D66, 163–175).

Author response: We thank the reviewer for her/his attention to this crystallographic pathology and for providing the useful references. We have added a short paragraph in the Methods section to more clearly explain the approaches used to determine the space group of this crystal. This section now also correctly refers to the crystal as pseudo-merohedrally twinned, provides the twin fraction, and includes a reference to a new supplemental figure (Supplementary Fig. 17) showing the L-test and H-test:

“F-VHH-4 crystals formed in space group $P2_12_12_1$ with near-perfect pseudo-merohedral twinning to appear as space group $P4_12_12$. Diffraction data from F-VHH-4 crystals were initially processed as $P422$ and a molecular replacement solution containing one VHH per asymmetric unit in space group $P4_12_12$ was obtained using PHASER⁴⁰. However, refinement resulted in suspiciously high $R_{\text{work}}/R_{\text{free}}$ values, particularly given the high resolution of this data set. Evaluation of systematic absences revealed that $00l$ reflections were present for $l=2n$, which is compatible with a 4_2 or 2_1 screw axis, but not with the 4_1 axis found in the molecular replacement solution. Evaluation of the L- and H-tests revealed that the crystal was pseudo-merohedrally twinned (Supplementary Fig. 17a) with a twin fraction greater than 0.4 (Supplementary Fig. 17b). The data were subsequently reprocessed in space group $P2_12_12_1$ and a molecular replacement solution was obtained containing two VHH molecules per asymmetric unit. After manual model building in Coot⁴¹, the structure was refined in PHENIX⁴² to an $R_{\text{work}}/R_{\text{free}}$ of 17.0/20.8% using twin law $kh-l$ and a refined twin fraction of 0.47.” Lines 388-401.

p. 18, Only the first letter in ‘Coot’ should be in caps, ‘PHENIX’ should be in all caps- these appear in multiple formats in the methods section.

Author response: The Methods section has been updated with the correct capitalization.

In Supplementary Figure 2, the wells that are boxed look like the other wells to me- are we supposed to see something here?

Author response: We apologize that the meaning of the boxed wells in supplementary figure 2 was not entirely clear. We have clarified this as follows in the legend of this figure:

“Boxes indicate *P. pastoris* clones with neutralizing activity (*i.e.* no RSV plaques or a reduced number of RSV plaques in one or more wells compared to the other clones).” Supplementary information lines 18-20.

In Supp. Figure 6, should there be standard deviation markers on the bars?

Author response: Supplementary Figure 6 shows the results of flow cytometry experiments that were performed to document binding of F-VHH-4 and -L66 to cells that were transfected with a RSV F expression construct (a) or to RSV infected cells (b). Each experiment was performed only once, hence standard deviations were not indicated. These binding experiments have now been repeated and a new Supplementary Figure 6 has been made based on the obtained results. Averages with standard deviations are now included based on three repeat experiments. The legend of Supplementary Figure 7 (new numbering in revised) has been adapted accordingly.

In regards to Supp. Figure 8, here or in the text, please report the RMSD between the bound and unbound VHH-4 structures.

Author response: The last part of the legend of Supplemental Figure 9 (previously numbered Supplementary Figure 8) now reads “The conformations of the two states are very similar, with an RMSD of 0.7 Å for 125 equivalent C α atoms.”

While VHH-4 loses some affinity binding monomeric F as compared to trimeric F, VHH-L66 does not bind the monomeric protein at all. Can the authors speculate as to the differences in affinity for the monomer, based on the crystal structures?

Author response: It is not entirely clear from the structures as to why VHH-4 and VHH-L66 differ in their binding to monomeric prefusion F, given that they bury identical surface areas on prefusion F. However, F-VHH-4 makes additional contacts that are not observed for F-VHH-L66 (Supplementary Table 4), including hydrogen bonds to both the β 3/ β 4 hairpin (RSV F Ser186 to VHH Tyr100F) and the loop connecting β 9 and β 10 (RSV F Asn345 to VHH Ser98). Although these contacts may help to stabilize the interaction of VHH-4 with monomeric prefusion F, we have decided to omit this analysis due to its speculative nature.

In Supp. Table 2, for the 1.87A structure, more reflections were used in refinement than were collected, and for the other two structures the numbers of reflections also differ by quite a lot- please check

these numbers carefully. Also there should be a comment somewhere in the text or table about the identity of the ligand/ion for the two low-resolution structures.

Author response: We thank the reviewer for careful evaluation of the crystallographic statistics. For all three structures, the total number of unique reflections collected is slightly higher than the total number of reflections used in refinement, as is often the case. Furthermore, the resolution shells used during data processing in AIMLESS are not identical to those used for refinement in PHENIX. For the 1.87 Å structure, the highest resolution shell assigned in PHENIX (1.94–1.87) is larger than the highest resolution shell assigned in AIMLESS (1.92–1.87).

We have also updated supplementary table 2 to reflect minor changes made to the structures during PDB validation and deposition, and provided the identity of the phosphate ions.

Reviewer #2 (Remarks to the Author):

The manuscript by Saelens et al describes the generation and isolation of a single domain antibody from llama that is highly potent in its ability to neutralize RSV A and B strains. The authors map the epitope of its binding on the F protein using escape mutation and X-ray crystallography, and show the epitope to be present on pre-F form of the fusion protein and binding to two adjacent protomers. Furthermore they demonstrate in vivo efficacy of the nanobody via delivering it intra-nasally into mice followed by RSV challenge. Overall based on their results the author's claim that their antibody could be used in humans to prevent RSV infection prophylactically. The manuscript could benefit from clearly laid out objective discussion as suggested below;

1) The in vitro potency of the antibody is derived from the plaque assay and is very much dependent on the input PFU. Many scientists in RSV research use 50-100 pfu as input virus for their experiments. However the current manuscript utilizes as low as 25-30 pfu only which could lead to lower or "ultra-potent" IC50's. In defense of the authors they do have comparator antibodies in the same assay and their nanobody is an order more potent than D25. It would be important to include the input virus pfu upfront in the figure and discuss the caveat of very low potency explained in parts due to use of lower viral pfu in their assays for neutralizations.

Author response: We agree with the reviewer that a two- to three-fold higher input inoculum will likewise require a two- to three-fold higher amount of nanobody (VHH) or conventional monoclonal antibody. In Figure 1 d-f we indeed show a direct comparison of F-VHH-4 and F-VHH-L66 with previously reported, potentially neutralizing monoclonal antibodies AM22 and D25 as well as with motavizumab and palivizumab. In Supplementary table S3, we had also included monoclonal antibodies MPE8, 101F and motavizumab as positive controls for neutralization.

We have also performed additional neutralization assays to compare the potency of the VHHs with conventional monoclonal antibodies against clinical RSV isolates where we included RSV A2 as a control using an input virus dose of 60 pfu/well. These data are shown in the new Supplementary Figure 4.

The input virus plaque forming units (PFU) is now included in the legend of Figure 1: Fig1a: 50 pfu/well; Fig1b: 40 pfu/well; Fig1d: 30 pfu/well; Fig1f: 25 pfu/well, and in the legend of the new Supplementary Figure 4: RSV A2 (60 pfu/well).

2) The table in Figure 2b does not match well with the figure 5A published earlier by Ngwuta and Graham et al (Sci Transl Med. 2015 October 14; 7(309): 309ra162. doi:10.1126/scitranslmed.aac4241.). Specifically In earlier manuscript 101F seems to compete with MPE8 (60-80% competition) but in the current manuscript it is much lower and more discrepant between the two modes of competition set up (4% and 40%). Furthermore, competition between 101F and AM14 is about 57% or 91% depending on the how the competition is done, while in the 2015 manuscript there was no difference - 85% both ways. Since Dr. Graham is also an author on the current manuscript the authors should be able to reconcile the differences and address them in the manuscript.

Author response:

As the reviewer points out, there are differences in antibody competition profiles between Ngwuta et al. and the present manuscript, especially with antibody 101F. These differences are attributed to how the assays were performed. In the Ngwuta et al. experiments, the Ds-Cav1 (prefusion F) protein was immobilized on biosensors through anti-His antibody, hence in a fixed orientation. In the present manuscript, because His-tags were also present on the nanobodies, the DS-Cav1 protein was immobilized on biosensors through amine coupling reaction, hence in a random orientation. The difference in protein orientation impacts the accessibility of some epitopes, especially sites that are sterically hindered by antibodies bound at a close proximity or by neighbouring protein and/or antibody bound to neighbouring proteins. This type of steric hindrance is more obvious in a fixed orientation, and reduced in a random orientation, particularly for epitopes on the membrane proximal aspect of the pre-F molecule like the 101F site IV. Based on our data and knowledge of the mAb epitope specificity, we believe both data in Ngwuta et al. and in the present manuscript are correct. To further clarify the differences in assays performed in the present manuscript, we revised associated Methods section as follows (revised lines 341-354):

“Antibody cross-competition was performed as described previously (Ngwuta et al.) with some modifications. Briefly, DS-Cav1 protein (10 µg/mL) was immobilized in a random orientation on AR2G biosensors (ForteBio) pre-activated with N-hydroxysuccinimide (NHS) and 1-ethyl-3-(3-dimethylaminopropyl)carbodiimide (EDC) through amine coupling reaction in acetate buffer (pH 5.0). The reaction was quenched by 1 M ethanolamine (pH 8.0) and DS-Cav1-immobilized biosensors were then equilibrated with assay buffer (PBS with 1% BSA). The biosensors were dipped in competitor antibodies (35 µg/mL in assay buffer) for 300 s followed by analyte antibodies (35 µg/mL in assay buffer) for 300 s with a short baseline step (60 s) in between the two antibody steps. All assays were performed at a set temperature of 30°C with agitation of 1,000 rpm in an Octet HTX instrument (ForteBio). Percent inhibition of antibody binding by competing mAbs was calculated with the following equation: Inhibition

(%) = 100 - [(analyte antibody binding in the presence competitor mAb) / (analyte antibody binding in the presence of isotype control mAb)] x 100.”

3) The authors suggest that the nanobody could be developed as a therapeutic for humans. It has been established using several antibodies (Palivizumab, Motavizumab, D25/MEDI8897 and REGN2222) that the cotton rat model is more useful in translating animal studies and amounts of antibody required for efficacy in humans. However, the authors chose to use the mouse model for their in vivo studies. Therefore it would be interesting to see the in vivo experiments be conducted in cotton rats to ascertain human clinical doses and comparison to ALX-0171

Author response: This is a valid suggestion, given that cotton rats are slightly more permissive for RSV replication than laboratory mice. Demonstrating inhibition of RSV replication in a second animal model will be an asset to further support the clinical development of the VHHs. However, a challenge experiment in cotton rats would require considerable time and efforts; none of the labs of the authors has this animal model in house. In addition, we anticipate that the outcome of such an experiment will likely only slightly increase the scientific content of the manuscript, given that the main findings reported in our manuscript are the strong antiviral effect of the single-domain antibodies that we discovered and their unique epitope recognition. It is also known that the virus neutralization activity that is needed to prevent RSV infection in cotton rats is very similar to the levels needed to prevent RSV infection of the lungs in mice (for which we show clear protection). Finally, when the single-domain antibodies that we describe move forward into clinical development, which we hope they will in the future, cotton rat studies will likely be done as part of that development.

ALX-0171 is a trimeric version of a nanobody that binds to the prefusion and postfusion form of RSV F and competes with palivizumab. ALX-0171 is being clinically developed by Ablynx Inc. Therefore, the binding properties, the size and the design of F-VHH-4/F-VHH-L66 are different from ALX-0171, which makes it difficult to make firm conclusions on the proposed comparison. Based on the publically available information on ALX-0171, we predict that the two VHHs described in our study would be better than ALX-0171 at neutralizing RSV B viruses.

4) Another minor point from a clinical development potential of VHH antibody is the disulfide bond found in CDR's which could turn out to be a manufacturing liability

Author response: This is a very interesting remark that relates to a fundamental requirement for the development of a biological to a possible product on the market, *i.e.* the possibility to produce high quantities with batch-to-batch consistency in terms of stability and biological activity. There is a higher chance that a single-domain antibody with a disulfide bond between two CDRs will express poorly or give rise to aggregates due to inappropriate disulfide formation compared to a single-domain antibody that lacks such a disulfide bond. However, this does not mean that all VHHs with a disulfide bond express poorly. While screening through the F-binding and RSV-neutralizing candidates produced in *Pichia pastoris* (secreted in the medium), we found no evidence that VHHs that lacked the predicted disulfide bond expressed better than VHHs with the disulfide bond (Coomassie-stained SDS-PAGE gel

shown below). We note that for some candidates the *Pichia pastoris* production was performed starting from two independent transformants, hence some of the lanes are labeled with the same number.

For example, like F-VHH-L66, VHHs nr. 8 and 44 have an Ala residue instead of a Cys residue at position 50 whereas candidates nrs. 1, 4 (= F-VHH-4) and 13 have a Cys residue at position 50:

```
F-VHH-8      QVQLQESGGGLVQPGGSLRLSVCNSGFTFSSYVMGWFRQAPGKERELVAAISGSG----TSKYAAPGEGRF
F-VHH-44    QVQLQESGGGLVQAGGSLRLSCAASGRTFSSYAMGWFRQAPGKREFVAAIKWSG----GSTYYADSVKGRF
F-VHH-13    QVQLQESGGGLVQAGGSLRLSCAASGFTFDDYAIGWFRQAPGKEREGVSCISSRE----GSTYYADSVKGRF
F-VHH-1     QVQLQESGGGLVQAGGSLRLSCAASGFTFDDYLIGWFRQAPGKEREGVSCISSSDGSTDGSTYYADSVKGRF
F-VHH-4     QVQLQESGGGLVQPGGSLRLSCAASGFTLDYYIGWFRQAPGKEREAVSCISGSS----GSTYYPDSVKGRF
F-VHH-L66   QVQLQESGGGLVQPGGSLRLSCAASGFTLDYYIGWFRQAPGKEREGVSCISSSSH----GSTYYADSVKGRF
```

```
F-VHH-8      SISRDNAKNTVSLAMSSLKPEDTAVYYCAATLDM-----GIYSGAYDYWGQGTQVTVSS
F-VHH-44    TISRDNAKNTVYLQMNSLKPEDTAVYYCAQQVYSDY---VQEVFVSQYEYDYWGQGTQVTVSS
F-VHH-13    TISSDSAKNTVYLQMNSLKPEDTAVYYCGASAQYGSTWYGLTGTCGYXXMNYWGKGTQVTVSS
F-VHH-1     TISSDNKNTVYLQMNSLKPEDTAVYYCAAIALT-----YYRGCLGGGMDYWGKGTQVTVSS
F-VHH-4     TISRDNAKNTVYLQMNSLKPEDTAVYYCATIRSS-----SWGCVHYGMDYWGKGTQVTVSS
F-VHH-L66   TISRDNAKNTVYLQMNSLKPEDTAVYYCATVAVA-----HFRGCGVDGMDYWGKGTQVTVSS
```

5) The most interesting data are on how the antibody binds to the pre-fusion F of RSV. These studies reveal that the antibody binds in a unique manner so far not seen with the human antibodies isolated and reported. The antibody binds to two adjacent protomer in a cavity with Thr50 in the pit of the cavity. This binding is similar to what has been reported for MPE8 (a siteIII binding human antibody) yet different that it engages additional surrounding residues not conserved with hMPV F protein (and therefore cannot neutralize hMPV). Furthermore it competes with Site II (Palivizumab), SiteIV (101F) and SiteV (AM14) binding antibodies. One interesting possibility is that due to the nanobody nature of VHH it may be able to access the cavity at the interface of two protomers and hence no human antibody (due to the size of the IgG format) is found that bind in this unique fashion. It is important to note that some site III human antibodies are more potent in neutralizing RSV as Fab than as full length antibodies (unpublished data). Therefore it would be of interest for the readers to see data of a monovalent Fab of MPE8 compared to full length MPE8 and VHH in an in vitro neutralization assay.

Author response: We thank the reviewer for pointing out the novelty of our findings. Although we agree that it would be interesting to compare the neutralization potency of the nanobodies to IgG and Fab

fragments of MPE8, the rationale for performing this experiment would not be obvious to the reader since it requires knowledge of the unpublished data referred to by the reviewer. Therefore, we have decided to omit such an experiment from the manuscript.

6) *The readers could also benefit from a broader survey of neutralization potency on actual clinical isolates of both RSV A and B strains.*

Author response: This is a valid remark that is in part covered by the data presented in Supplementary Table S1, where we summarize the neutralization potency of a set of recombinant RSV viruses that express F and G derived from primary clinical isolates. We have now also tested the potency of F-VHH-4, F-VHH-L66, palivizumab, motavizumab, D25 and AM22 against the following clinical isolates: MAD/GM2_2/12, MAD/GM2_12/12, MAD/GM2_13/12, MAD/GM2_14/12, MAD/GM3_10/14, MON/9/92 (primary RSV A isolates) and MAD/GM3_7/13 (a primary RSV B isolate). In all cases the IC₅₀ concentration of F-VHH-4 and F-VHH-L66 was lower than that of any of the monoclonal antibodies evaluated in parallel (New supplementary Figure 4) .

Reviewer #3 (Remarks to the Author):

This manuscript describes the generation of llama single-domain monoclonal antibodies specific for respiratory syncytial virus pre-fusion F protein and their characterization. The data presented include detailed presentation of the methods of generation of the antibodies as well as detailed structural studies of the antibodies alone or in a complex with the pre-fusion F protein and analyses of the biological activities of the antibodies both in tissue culture and in animals.

The manuscript is clearly written, very interesting, and describes development of two novel antibodies that define a new antigenic site on the pre-fusion F protein, albeit a site that is not likely accessed by human antibodies as indicated by the authors' structural analysis. The results of in vitro and animal studies strongly support the authors' proposal that these antibodies have significant potential as prophylaxis or possible treatment of RSV induced disease. These antibodies also have significant potential as reagents for basic studies of virus infection, including virus entry and refolding of the pre-fusion F to the post fusion form. Indeed their analysis indicates that inhibition of infection by these antibodies may be due to stabilization of the trimer structure of the pre-fusion F protein. In general, the data are very comprehensive and the quality of results is mostly excellent.

There are several novel aspects of the study. First, the authors have defined a new antigenic site accessed only by single domain antibodies. Second, they describe a novel potential prophylactic or therapeutic agent unlike any previously tested for RSV. Third, they describe a new reagent for basic studies of virus infection.

The authors do describe the generation of the antibodies in detail. It is not clear that such detail is necessary since protocols are moderately widely available and offered by numerous companies.

There are some minor problems that should be addressed.

Author response: We thank the reviewer for the appreciation of our work.

1. Page 3, line 20: a reference for antibody preventing F remodeling needs to be added.

Author response: We found it very hard to identify a primary reference that demonstrates such a mechanism for F-specific antibodies. Therefore, we have changed the sentence in line 51 on page 3 as follows:

“Small molecules that bind to RSV F and prevent its structural remodeling or F-specific antibodies that interfere with membrane fusion can block RSV infection^{7,8}.”

2. Page 4, line 11: references for the use of VHHs antibodies for treatment of rheumatoid arthritis and cancer should be added.

Author response: These references have been added.

3. Page 10, lines 5-8, supplementary fig 10: a description of the generation of monomer F protein needs to be added to materials and methods or at least a reference for preparation of this form should be added.

Author response: We have added a description of the monomeric protein used to the figure legend of Supplementary Fig. 11 (previously Supplementary Fig. 10).

4. Page 11, line 13: text indicates lungs in Supplemental Fig 14 but in the figure the data are described as characterizing BAL. Which is it?

Author response: We apologize for this confusion. In Figure 6a viral loads in the lung homogenates are presented and in Supplementary Figure 14, virus titers in the BAL are depicted. This has now been clarified in the text (line 216).

5. Some of the Y-axes on figures are not clearly labeled: Figure 1a, 1b, 1d, 1f; Figure 3 both panels; Figure 6a, 6b, and 6c; Supplemental figure 4: Y-axis needs a label. Label on the side of the chart in Fig 2b is unclear.

Author response: Labeling of the Y axes has been improved.

6. *Supplemental figure 7: This is the only figure that is of questionable quality. The signals in panels in A, particularly the red signal, are very hard to see. This figure could be improved.*

Author response: The signals in Supplementary Fig. 7 (in the revised manuscript numbered Supplementary. Fig. 8) have now been improved by increasing the brightness of the red fluorescent signal of the panels displaying the goat anti-RSV immunofluorescent staining.

REVIEWERS' COMMENTS:

Reviewer #1:

[No further comments for author.]

Reviewer #2 (Remarks to the Author):

The revised manuscript is much improved and acceptable for publication.

It would be important to provide accession numbers or F sequences for the MAD/GM2_2/12, MAD/GM2_12/12, MAD/GM2_13/12, MAD/GM2_14/12, MAD/GM3_10/14, MON/9/92 (primary RSV A isolates) and MAD/GM3_7/13 (a primary RSV B isolate) clinical isolates

Reviewer #3 (Remarks to the Author):

The authors have provided appropriate responses to all comments made by reviewers of the first version of this manuscript. This excellent manuscript is now improved and results are clarified.

Reviewer #1:

[No further comments for author.]

Reviewer #2 (Remarks to the Author):

The revised manuscript is much improved and acceptable for publication.

Author response: We thank the reviewer for this feedback and advice.

It would be important to provide accession numbers or F sequences for the MAD/GM2_2/12, MAD/GM2_12/12, MAD/GM2_13/12, MAD/GM2_14/12, MAD/GM3_10/14, MON/9/92 (primary RSV A isolates) and MAD/GM3_7/13 (a primary RSV B isolate) clinical isolates

Author response: This is a good suggestion. The G protein gene sequences of these clinical isolates have been reported before, which revealed that these viruses represent at least 2 different genotypes of antigenic group A (GA1 and GA2) and one antigenic group B [Trento et al., J. Virol., 89:7776-7785 (2015) and García et al., J. Virol., 68:5448-5459 (1994)]. However, the F gene sequences of these viruses have not yet been determined. We note that the epitope that is recognized by F-VHH-4 and F-VHH-L66 is conserved in F. The *in vitro* neutralization data also indicate that these viruses are susceptible to neutralization by previously published conventional monoclonal antibodies (Palivizumab, Motavizumab, AM22 and D25). The prefusion F-specific mAbs neutralize these viruses better than Pali- and Motavizumab do, which is in line with reports that have been published by other groups. Therefore, in our opinion, it is not essential to include the F sequences of these viruses in our manuscript. However, we have started this F sequence analysis and hope to be able to publish the obtained sequences in the coming months as directly submitted Genbank entries.

Reviewer #3 (Remarks to the Author):

The authors have provided appropriate responses to all comments made by reviewers of the first version of this manuscript. This excellent manuscript is now improved and results are clarified.

Author response: We thank the reviewer for her/his appreciation of our work.